# Evaluating seasonal sea-ice cover over the Southern Ocean at the Last Glacial Maximum

Ryan A. Green[1,2,3], Laurie Menviel[1,4], Katrin J. Meissner[1,2], Xavier Crosta[5], Deepak Chandan[6], Gerrit Lohmann[7,8], W. Richard Peltier[6], Xiaoxu Shi[7], and Jiang Zhu[9]

[1]Climate Change Research Centre, University of New South Wales, Sydney, Australia
[2]ARC Centre of Excellence for Climate System Science, Sydney, Australia
[3]Earth and Planetary Sciences, University of California, Santa Cruz, USA
[4]The Australian Centre for Excellence in Antarctic Science, University of Tasmania, Hobart, Tasmania 7001, Australia
[5]Université de Bordeaux EPOC, UMR 5805, Pessac, France
[6]Department of Physics, University of Toronto, 60 St. George Street, Toronto, Ontario, M5S 1A7, Canada
[7]Alfred Wegener Institute, Helmholtz-Zentrum für Polar- und Meeresforschung, Bremerhaven, Germany
[8]Institute for Environmental Physics, University of Bremen, Bremen, Germany
[9]Climate and Global Dynamics Laboratory, National Center for Atmospheric Research, Boulder, Colorado, USA

**Correspondence:** Ryan A. Green (rygreen@ucsc.edu)

**Abstract.** Southern hemispheric sea ice impacts ocean circulation and the carbon exchange between the atmosphere and the ocean. Sea ice is therefore one of the key processes in past and future climate change and variability. As climate models are the only tool available to project future climate change, it is important to assess their performance against observations for a range of different climate states. The Last Glacial Maximum (LGM, ~21,000 years ago) represents an interesting target as it

is a relatively well documented period with climatic conditions very different from pre-industrial conditions. Here, we analyse the LGM seasonal Southern Ocean sea-ice cover as simulated in numerical simulations part of the Paleoclimate Modelling Intercomparison (PMIP) phases 3 and 4. We compare the model outputs to a recently updated compilation of LGM seasonal Southern Ocean sea-ice cover and summer sea surface temperature (SST) to assess the most likely LGM Southern Ocean state. Simulations and paleo-proxy records suggest a fairly well-constrained glacial winter sea-ice edge between 50.5˚ and

51˚S. However, the spread in simulated glacial summer sea ice is wide, ranging from almost ice-free conditions to a sea-ice edge reaching 53˚S. Combining model outputs and proxy data, we estimate a likely LGM summer sea-ice edge between 61˚ and 62˚S and a mean summer sea-ice extent of 14-15 x $10^6$ km$^2$, which is ~20-30% larger than previous estimates. These estimates point to a higher seasonality of southern hemispheric sea-ice during the LGM than today. We also analyse the main processes defining the summer sea-ice edge within each of the models. We find that summer sea-ice cover is mainly defined

by thermodynamic effects in some models, while the sea-ice edge is defined by the position of Southern Ocean upwelling in others. For models included in both PMIP3 and PMIP4, this thermodynamic or dynamic control on sea ice is consistent across both experiments. Finally, we find that the impact of changes in large-scale ocean circulation on summer sea ice within a single model is smaller than the natural range of summer sea-ice cover across the models considered here. This indicates that care must be taken when using a single model to reconstruct past climate regimes.

# 1  Introduction

Antarctic sea ice plays an important role in the Earth's climate system, affecting marine productivity, air-sea gas exchange, air-sea heat fluxes, surface water density, and surface albedo. It can both impact and respond to changes in bottom-water formation and Southern Ocean (SO) circulation, and therewith impact large-scale heat transport and ocean carbon uptake. While Arctic sea-ice cover has significantly decreased over the last few decades, Antarctic sea-ice cover has been more dynamic. It slowly expanded from the late 1970's until 2016, and then sharply declined for the next three years (Eayrs et al., 2021). Doddridge and Marshall (2017) have shown that, on a seasonal timescale, SO sea ice was responding to changes in the southern annual mode (SAM), with a positive phase of the SAM leading to lower SO sea surface temperatures (SST) and a larger sea-ice extent. However, on longer timescales, a positive phase of the SAM can lead to a SO warming due to the enhanced upwelling of relatively warm circumpolar deep waters (Ferreira et al., 2015). Due to continued anthropogenic emissions of carbon dioxide, the southern hemispheric westerly winds are projected to strengthen and to shift towards positive phases of the SAM (Zheng et al., 2013), impacting SO circulation and sea-ice cover further (Mayewski et al., 2017). Given that the SO has accounted for ∼40% of the oceanic anthropogenic $CO_2$ uptake between 1870 and 1995 (Landschützer et al., 2015; Sabine et al., 2004; Frölicher et al., 2015; Mikaloff-Fletcher et al., 2006; Sabine et al., 2004; Watson et al., 2020), it is crucial to better understand the processes that impact Antarctic sea-ice cover. Specifically, seasonal changes in sea ice can significantly affect SO dynamics through buoyancy (Marzocchi and Jansen, 2017) and lead to changes in the atmosphere-ocean carbon exchange (Haumann et al., 2016). Understanding changes in sea ice and their natural drivers at different timescales and under different boundary conditions will allow us to better project future sea-ice changes.

The Last Glacial Maximum (LGM, ∼21,000 years ago) featured large continental ice-sheets over North America and Eurasia (e.g., Carlson and Winsor, 2012; Clark et al., 2009), as well as an extended Antarctic ice-sheet (Bentley et al., 2014), and an atmospheric $CO_2$ concentration of ∼185 ppm (Marcott et al., 2014). Despite significant progress in characterizing the LGM sea-surface conditions (e.g., Waelbroeck et al., 2009), oceanic circulation (e.g., Howe et al., 2016; Lynch-Stieglitz et al., 2007; Meissner et al., 2003; Menviel et al., 2017; Skinner et al., 2017), and mechanisms leading to a lower atmospheric $CO_2$ concentration (e.g., Kohfeld and Chase, 2017), significant uncertainties remain. Some of these uncertainties lie in characterizing seasonal Antarctic sea ice. While LGM Antarctic sea ice was first reconstructed in 1981 (CLIMAP-Project-Members, 1981), the proxy compilation of Gersonde et al. (2005) is nowadays routinely used to provide estimates of LGM sea-ice cover. Since 2005, additional SO sea-ice data has been published (Allen et al., 2011; Benz et al., 2016; Ferry et al., 2015; Ghadi et al., 2020; Nair et al., 2019; Xiao et al., 2016), and recently merged into an updated compilation (Lhardy et al., 2021). Within this updated sea-ice compilation, certain cores also contain summer SST estimates. We use this sea-ice proxy data along with the summer SST proxy data to better constrain the minimum and maximum LGM sea-ice cover.

Although paleo-proxy records are an invaluable tool to reconstruct the climate system, they are sometimes scarce or completely absent over entire regions. Climate models can help fill these gaps, as they provide a full 3-dimensional and dynamically consistent representation of the climate system. However, as climate models are not perfect representations of reality, it is important that we continually evaluate their performance. The Paleoclimate Intercomparison Project (PMIP) has been

set up to evaluate and compare model performances across consistent boundary conditions (Kageyama et al., 2017). Results from the PMIP phase 4 are currently being released (Kageyama et al., 2020), while phases 1-3 are available to the public (https://pmip3.lsce.ipsl.fr).

PMIP2 LGM simulations suggested that simulated LGM Antarctic sea-ice cover did not reflect the zonal variability nor the seasonality seen in proxy reconstructions (Roche et al., 2012). PMIP3 LGM simulations have also been analyzed, however not regionally, with results highlighting large inter-model differences in annual-mean, minimum and maximum Antarctic sea-ice area, and suggesting most PMIP3 models underestimate austral winter sea-ice cover in comparison to proxy data (Sime et al., 2016; Marzocchi and Jansen, 2017). Therefore, a regional analysis of seasonal LGM sea ice simulated by PMIP3 models is lacking. Furthermore, no seasonal sea-ice analysis of PMIP4 simulations under LGM boundary conditions has been performed yet.

Here, we assess the minimum and maximum SO sea-ice extent as simulated in LGM PMIP3 and PMIP4 experiments. To better assess intra- versus inter-model variability, a suite of LGM sensitivity experiments performed with the LOVECLIM model of intermediate complexity are also included. The PMIP3, PMIP4 and LOVECLIM experiments are compared to available sea-ice and SST paleo-proxy data, allowing us to determine the best model-data fit. Combining models and proxy data, we can provide an updated estimate of seasonal SO sea-ice cover during the LGM. Furthermore, we analyze the processes that lead to the inter-model differences in summer sea-ice extent at the LGM.

## 2   Methods

### 2.1   LGM numerical simulations

In this study, we include all PMIP3 and PMIP4 LGM simulations which provide sea-ice variables in the PMIP3 and PMIP4 database (Table 1). Each LGM simulation follows either PMIP3 or PMIP4 protocol (Kageyama et al., 2017; Braconnot and Kageyama, 2015). The PMIP3 protocol calls for all models to use the same ice sheet reconstruction, while the PMIP4 protocol allows for the use of either the original PMIP3 ice sheet to facilitate comparison with earlier simulations, or one of the newer reconstructions ICE-6G_C (Argus et al., 2014; Peltier et al., 2015) and GLAC-1D (Ivanovic et al., 2016). In total, data from eight models was obtained for PMIP3 and from six models for PMIP4 (Table 1). Three models submitted two different simulations to PMIP3 (CCSM4, GISS-E2-R, MPI-ESM-P). These simulations differed because of a difference in the initial state (CCSM4), or small changes in the physics of the model (GISS-E2-R and MPI-ESM-P). Following Sime et al. (2016), we chose to average the simulations for the models who submitted two LGM runs, yielding one output per model.

We also include three additional LGM simulations performed with the Earth system model of intermediate complexity, LOVECLIM (Goosse et al., 2010). LOVECLIM consists of an ocean general circulation model, a dynamic-thermodynamic sea-ice model, coupled to a quasi-geostrophic atmospheric model, a dynamic vegetation model and a carbon cycle model (Goosse et al., 2010). One of the LOVECLIM simulations follows the PMIP4 protocol, while two additional simulations were obtained by transiently forcing the model between 35 and 20 thousand of years before present with appropriate boundary conditions (i.e. orbital parameters, Northern Hemispheric ice-sheet topography and albedo) (Menviel et al., 2017). During

**Table 1.** Models analysed in this study, the PMIP phase they pertain to, and the ice-sheet forcing that was used. When applicable, the ensemble member (rip-PMIP3/ripf-PMIP4) for the model is specified. IcIES is the Ice sheet model for Integrated Earth system Studies forced by climatic outputs from MIROC (Abe-Ouchi et al., 2013). The atmospheric $CO_2$ concentration (p$CO_2$) for PMIP3 and PMIP4 experiments is 185 ppm and 190 ppm, respectively. p$CO_2$ for the two LOVECLIM sensitivity experiments is given.

| Model | Reference | Ice sheets | PMIP phase and rip(f) | Additional comments |
|---|---|---|---|---|
| CNRM-CM5 | Voldoire et al. 2013 | PMIP3 | PMIP3 r1i1p1 | |
| GISS-E2-R | Schmidt et al. 2014, 2011; Ullman et al. 2014 | PMIP3 | PMIP3 r1i1p150/r1i1p151 | r1i1p150 - PMIP3-ice sheet / r1i1p151 - ICE-5G ice extent with lower Laurentide Ice Sheet altitude |
| IPSL-CM5A-LR | Dufresne et al. 2013; Kageyama et al. 2013 | PMIP3 | PMIP3 r1i1p1 | |
| MIROC-ESM-P | Sueyoshi et al. 2013; Watanabe et al. 2011 | PMIP3 | PMIP3 r1i1p1 | |
| MPI-ESM-P | Giorgetta et al. 2013; Klockmann et al. 2016 | PMIP3 | PMIP3 r1i1p1/r1i1p2 | r1i1p1 - AO / r1i1p2 - AOV and initial state spun up from PMIP2 simulations |
| MRI-CGCM3 | Yukimoto et al. 2012 | PMIP3 | PMIP3 r1i1p1 | |
| FGOALS-G2 | Li et al. 2013; Zheng and Yu 2013 | PMIP3 | PMIP3 r1i1p1 | |
| CCSM4 | Gent et al. 2011; Brady et al. 2013 | PMIP3 | PMIP3 r1i1p1/r2i1p1 | |
| MIROC-ES2L | Hajima et al. 2020 | ICE- 6G_C | PMIP4 r1i1p1f2 | |
| IPSL-CM5A2 | Sepulchre et al. 2020 | ICE- 6G_C | PMIP4 r1i1p1 | |
| MPI-ESM1.2 | Mauritsen et al. 2019 | ICE- 6G_C | PMIP4 r1i1p1f1 | |
| AWI-ESM-1 | Sidorenko et al. 2015 | ICE- 6G_C | PMIP4 | |
| CESM1.2 | Tierney et al. 2020 | ICE- 6G_C | PMIP4 | |
| UoT-CCSM4 | Chandan and Peltier 2017, 2018; Peltier and Vettoretti 2014 | ICE- 6G_C | PMIP4 | |
| LOVECLIM 1.2 | Goosse et al. 2010 | ICE-6G_C | PMIP4 | |
| LOVECLIM 1.2 weakNA | Goosse et al. 2010; Menviel et al. 2017 | IcIES | | p$CO_2$= 203 ppm, freshwater input into North Atlantic (0.05 Sv) |
| LOVECLIM 1.2 weakNA_AB | Goosse et al. 2010; Menviel et al. 2017 | IcIES | | p$CO_2$= 191 ppm, freshwater input to North Atlantic (0.05 Sv), Southern Ocean (0.1 Sv), and 20% weakening of southern hemispheric westerlies |

the 35 ka spinup the atmospheric $CO_2$ concentration was set at 190 ppm, after which $CO_2$ was a prognostic variable. In these simulations the oceanic circulation was altered by i) adding 0.05 Sv of freshwater to the North Atlantic to simulate a weaker and shallower North Atlantic Deep Water (NADW) formation at the LGM compared to pre-industrial (simulation V3LNAw in Menviel et al. 2017, here referred to as weakNA), ii) by adding 0.05 Sv of freshwater to the North Atlantic, 0.1 Sv to the SO, as well as by weakening the southern hemispheric westerlies by 20% to simulate a weaker LGM NADW and Antarctic Bottom Water (AABW) formation (simulation V3LNAwSOwSHWw in Menviel et al. 2017, here referred to as weakNA_AB). Atmospheric $CO_2$ was calculated prognostically in these experiments and equals 203 ppm in weakNA and 191 ppm in weakNA_AB, compared to 185 ppm in the PMIP3 protocol, and 190 ppm in the PMIP4 protocol. These two simulations will be referred to as the "LOVECLIM sensitivity runs". They were chosen because they provided the best model-data fit against a range of paleo-proxy records, including phosphate, $\delta^{13}C$, radiocarbon ventilation ages, and eps(Nd) (Menviel et al., 2017, 2020), thus indicating an appropriate oceanic circulation representation. The three LOVECLIM simulations can provide information on the impact of oceanic circulation differences on SO sea ice and SST, and thus allow us to assess intra- versus inter-model differences.

To ease the comparison, we used bilinear interpolation to standardize each model to a 1° x 1° grid with the CDO software (Climate Data Operators, Schulzweida et al. 2014).

## 2.2 Proxy data

The numerical simulations are compared to a compilation of 149 proxy records covering the LGM (see Table S1 in the Supplement, Allen et al. 2011; Benz et al. 2016; Ferry et al. 2015; Gersonde et al. 2005; Ghadi et al. 2020; Nair et al. 2019; Xiao et al. 2016). Quantitative SST was reconstructed at 138 locations, proxies for winter sea-ice presence or concentration were available at 149 locations and proxies for summer sea-ice presence were available at 132 locations. SSTs were derived from diatom-based transfer functions (Crosta et al., 1998; Esper and Gersonde, 2014a) while winter and summer sea-ice extent were derived either from the relative abundance of sea-ice indicator diatoms, respectively the *Fragilariopsis curta* group and *F. obliquecostata* (Gersonde et al., 2005), or diatom-based transfer functions whenever possible (Crosta et al., 1998; Esper et al., 2014b). Relative abundances of the indicator diatoms above 3% are thought to indicate the presence of sea ice over the core site (mean sea-ice extent north of the core site) while relative abundances between 1 and 3% suggest the episodic presence of sea ice over the core site (mean sea-ice edge south of the core site but maximum sea-ice edge north of the core site). In this study, we characterize the relative abundance of >3% as evidence of sea ice and the relative abundance between 1 and 3% as evidence for possible sea ice. Quantitative values were considered to indicate the presence of winter sea ice when they were above the root mean square error of prediction (RMSEP) on the validation models, generally around 10% for winter sea ice (Crosta et al., 1998; Esper et al., 2014b). Quantitative values were always below the RMSEP of ∼10% for summer sea ice in the validation model.

**Table 2.** Austral winter and austral summer months used for each simulation.

| PMIP3 models | Austral winter | Austral summer | PMIP4 models | Austral winter | Austral summer |
|---|---|---|---|---|---|
| MIROC-ESM-P | September-October | February-March | MIROC-ES2L | September-October | February-March |
| IPSL-CM5A-LR | August-September | February-March | IPSL-CM5A2 | August-September | February-March |
| MPI-ESM-P | September-October | February-March | MPI-ESM1.2 | September-October | February-March |
| CCSM4 | September-October | March-April | UoT-CCSM4 | September-October | March-April |
| CNRM-CM5 | September-October | February-March | CESM1.2 | August-September | February-March |
| GISS-E2-R | September-October | February-March | AWI-ESM-1 | August-September | February-March |
| MRI-CGCM3 | September-October | February-March | LOVECLIM | August-September | February-March |
| FGOALS-G2 | September-October | March-April | - | - | - |
| LOVECLIM sensitivity runs | Austral winter | Austral summer | | | |
| weakNA | July-August | February-March | - | - | - |
| weakNA_AB | August-September | February-March | - | - | - |

## 2.3 Definitions of sea-ice edge, extent, seasonality, and regions

We analyze the climatology of Antarctic sea-ice extent and define the two months of maximum and minimum sea ice for each
individual model (Table 2). These two months of maximum and minimum sea ice are used consistently throughout the study
and will hereafter be referred to as each model's austral "winter" and "summer", respectively. We note that using a two month
average leads to a larger summer and a smaller winter sea-ice extent, compared to what would be obtained from a one month
average. However, we believe that a two month average is more appropriate for a comparison with proxy records. We also
analyze simulated sea ice within specific regions which we refer to by the ocean basin the region lies in: Atlantic, Pacific, and
Indian Sector.

The sea-ice edge is defined as the 15% sea-ice concentration isoline. We calculate this by zonally averaging across all longi-
tudes for each latitude band, then determining at which latitude the model simulates a minimum of 15% sea-ice concentration.
For model simulations that do not reach 15% of sea-ice concentration in some regions of the SO, we average only over the
remaining regions with sufficient sea-ice cover. For model simulations that do not reach 15% sea-ice concentration in any
region, we define the latitude of their sea-ice edge as the latitude of the Antarctic coast. It is important to note that although a
model's sea-ice edge gives insight into its sea-ice characteristics, it is not always an accurate representation of how much total
sea ice a model simulates. Due to this, we also calculate the total sea-ice extent for each model (using a cut-off limit of 15% in
concentration).

To calculate the multi-model mean (MMM), we average sea-ice concentration over each grid cell for all models (PMIP3,
PMIP4, and LOVECLIM sensitivity runs separately). We then calculate the 15% sea-ice concentration isoline of each MMM.
To calculate the standard deviation, we similarly compute a standard deviation value for each individual grid cell, before

adding and subtracting that standard deviation ($\sigma$) of sea-ice concentration from the MMM for each grid cell. The +/- 1$\sigma$ then represents the 15% sea-ice concentration isoline calculated from the MMM +/- 1$\sigma$. Notably, this creates a non-symmetric standard deviation isoline as each grid cell has its own MMM (and $\sigma$) value, calculated independently from any surrounding
grid cells.

## 3 Results

Figure 1 shows the austral winter and austral summer mean LGM sea-ice extent as simulated by each model considered here as well as the MMM and one standard deviation for the PMIP3, PMIP4 and LOVECLIM models. For comparison, available paleo-proxy records are overlaid for austral winter and austral summer. The simulated annual mean LGM sea-ice extent is
145 shown for all the models in Fig. S1.

### 3.1 Simulated austral winter SO sea-ice extent and comparison with winter sea-ice proxy records

During austral winter, the simulated zonally averaged MMM sea-ice edge for PMIP3 models lies at ∼51.5˚S with one standard deviation equating to 1.5˚ north and 5˚ south of the MMM (Fig. 1d, Table 3). Regional differences are found across the models. The standard deviation in the Indian Ocean sector (20˚E to 147˚E) increases to the south due to GISS-E2-R (cyan) simulating a
150 sea-ice edge reaching 59.5˚S in that sector, compared to the Indian sector MMM of 50.5˚S. The standard deviation south of the MMM is also large in the Pacific sector (147˚E to 68˚W) as CNRM-CM5 (black) simulates a sea-ice edge at ∼59.5˚S compared to the Pacific sector MMM of ∼56.5˚S. There is higher model agreement in the Atlantic sector (68˚W to 20˚E) leading to a standard deviation of only +0.5/-1.5˚ in that region.

Similar to PMIP3 models, PMIP4 models simulate a MMM sea-ice edge at ∼51˚S during austral winter with a zonally
averaged standard deviation of 2˚ north and 5˚ south of the MMM (Fig. 1h, Table 2). UoT-CCSM4 (orange) simulates the largest sea-ice cover with a sea-ice edge at ∼48˚S. Conversely, MIROC-ES2L (purple) simulates significantly less sea ice than all other models, despite having a slightly extended sea-ice edge north of 60˚S in the Atlantic sector. IPSL-CM5A2 (green) also simulates limited sea-ice cover in the Pacific sector. Due to MIROC-ES2L's extended sea-ice edge in the Atlantic and a relatively good agreement between the other six models in that region, the standard deviation is smallest in the Atlantic sector
with values of +1˚/ -3˚.

During austral winter both LOVECLIM sensitivity runs simulate similar sea-ice cover despite the different circulation forcing. The MMM sea-ice edge is simulated at ∼51.5˚S with a standard deviation of 0.5˚ north and 1˚ south. PMIP3, PMIP4 and LOVECLIM models thus simulate similar MMM sea-ice edge locations between 51˚ and 51.5˚S. There are slight regional differences due to a few models displaying a different sea-ice edge in a particular sector, however the zonally averaged standard
deviations produce similar values for PMIP3 and PMIP4. With only two similar models included in the LOVECLIM sensitivity MMM, the standard deviation is much smaller.

We next compare the simulations with proxy data of sea-ice presence or absence. A relative abundance of *F. obliquecostata* greater than 3% (Fig. 1 - blue filled circles) indicates the presence of sea ice, relative abundances of *F. obliquecostata* between

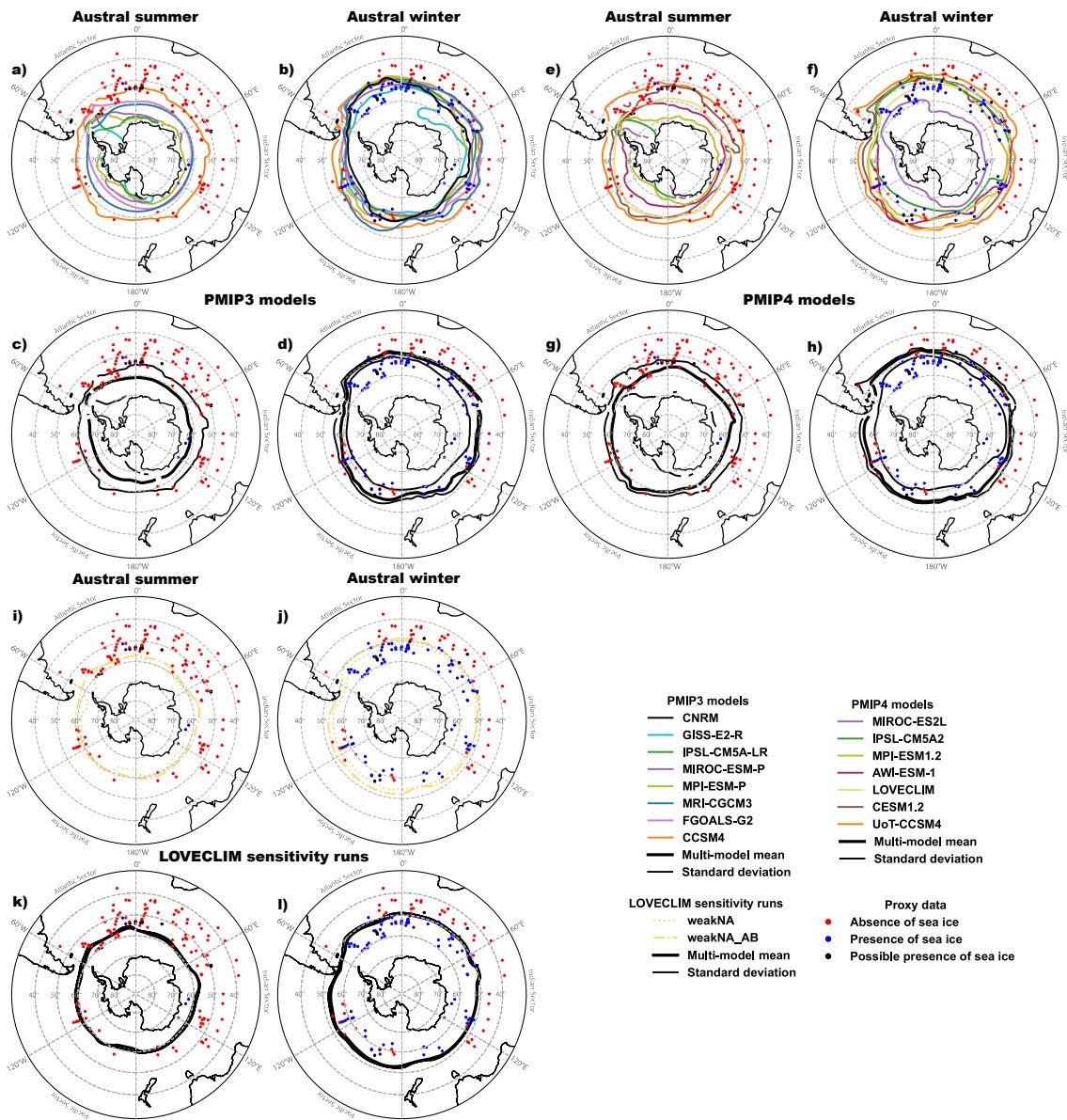

**Figure 1.** Simulated summer and winter sea-ice edge compared to proxy data for PMIP3 (a-d) and PMIP4 (e-h) models as well as LOVE-CLIM sensitivity runs (i-l). The left column for each group shows simulated austral summer sea-ice concentration at 15% (a, c, e, g, i, k) and the right column shows simulated austral winter sea-ice concentration at 15% (b, d, f, h, j, l). The top row for each group shows individual model results (a, b, e, f, i, j) while the bottom row shows the multi-model mean ± one standard deviation (c, d, g, h, k, l). Blue, black, and red filled points represent the sediment core proxy data used for the study.

1% and 3% suggest the possible presence of sea ice (Fig. 1 - black filled circles), and core locations with a relative abundance

of *F. obliquecostata* < 1% indicate ice-free conditions (Fig. 1 - red filled circles). We use these data points to calculate a model-data sea-ice agreement percentage, based on whether the model simulation correctly or incorrectly simulates the sea-ice state at the location of the proxy data. For these calculations, we characterize the possible presence of sea ice (1-3% [*F. obliquecostata*]) as ice-free.

All three MMMs for austral winter display a model-data sea-ice agreement between 83 and 84% (Fig. 1, Table 3). Four proxy data points indicate the presence of sea ice extending past 50˚S, all located within the Atlantic sector and into the western-most edge of the Indian sector. Both the PMIP3 and PMIP4 MMMs simulate this feature as their sea-ice edge extends equatorward of 50˚S within the Atlantic sector. In regard to individual model performance, FGOALS-G2 (pink) simulates the highest model-data sea-ice agreement with 87%, corresponding to a simulated sea-ice edge of 50.5˚S. With an extended sea-ice edge in the Atlantic and slightly retreated sea-ice edge within the Pacific, FGOALS-G2 seems to most accurately simulate the regional distribution of the proxy data. The data and modelling constraints thus suggest that the LGM austral winter sea-ice (WSI) edge was likely between 50.5˚S and 51˚S and the mean LGM austral WSI extent was likely around 35-36 x $10^6$ km$^2$ .

## 3.2   Simulated austral summer SO sea-ice extent and comparison with summer sea-ice proxy records

A larger spread among the models is obtained during austral summer (Fig. 1a, c), with a MMM sea-ice edge at ∼62.5˚S and a zonally averaged standard deviation of 5˚ north and 11˚ south of the PMIP3 MMM. The largest sea-ice cover is simulated by CCSM4 (orange) with a sea-ice edge at ∼55.5˚S. Three models (CNRM-CM5, black; GISS-E2-R, cyan; IPSL-CM5A-LR, green) only simulate sea ice around the Ross and Weddell Seas and are otherwise ice-free. CNRM-CM5 (black) simulates the least amount of sea ice at or above 15% concentration with sea ice only simulated in a small region of the Ross Sea (Fig. 1a).

The austral summer MMM in PMIP4 models is larger than for PMIP3 with a sea-ice edge at ∼59.5˚S and a zonally averaged standard deviation of 4.5˚ north and 10.5˚ south of the MMM (Fig. 1g, Table 2). Similar to austral winter, UoT-CCSM4 (orange) simulates the largest sea-ice cover with a sea-ice edge reaching ∼53˚S while MIROC-ES2L (purple) and IPSL-CM5A (green) simulate the least amount of sea ice with sea ice only found over the Ross and Weddell Seas.

There are slightly larger sea-ice differences between the two LOVECLIM sensitivity runs during austral summer than winter. The MMM sea-ice edge is simulated at ∼59˚S with a standard deviation of 0.5˚ north and 1˚ south of the MMM. WeakNA (dotted yellow) has more regional variability than weakNA_AB (dash-dot yellow). Due to the standard deviation being so small for both austral winter and austral summer, the lines are difficult to distinguish from the MMM. The simulated austral summer sea-ice (SSI) edge is thus more poorly constrained than the winter sea-ice edge across the different groups, with a MMM of 59˚ to 62.5˚S, and large standard deviations of 5˚ north and 11˚ south of the mean.

The simulated summer sea-ice extent can also be compared to paleo-proxy records. Only six core locations out of 132 indicate the presence of SSI while seven additional cores from the Atlantic sector of the SO suggest the possible presence of SSI. The remaining 119 core locations indicate ice-free conditions (including 64 cores with 0% relative abundance of *F. obliquecostata*). Of the six locations indicating the presence of sea ice, three cores are located in the Indian sector at ∼63˚S south of the MMM for all three model groups, whereas the other three are located in the Atlantic sector at ∼53˚S, north of the MMM for all three model groups (Fig. 1). However, two of the three cores within the Atlantic Sector indicating LGM SSI fall

inside the PMIP4 +1 standard deviation contour line (Fig. 1g). Five of the seven locations indicating the possible presence of
SSI are located north of all three MMMs, but again on or inside the PMIP4 +1 standard deviation (Fig. 1g). We note that the
eight locations from the Atlantic sector representing a presence or possible presence of SSI are bordered by cores suggesting
ice-free conditions, possibly indicating a sea-ice tongue protruding from the Weddell Sea. The reader should bear in mind that
with limited proxy data points indicating SSI and multiple models simulating limited SSI, it is not uncommon to record the
same summer model-data sea-ice agreement percentage across different model simulations.

With only six proxy data points indicating SSI, the PMIP3 MMM, which displays the smallest SSI extent across MMMs,
shows the highest model-data sea-ice agreement (Table 3). In terms of individual models, it is clear some models simulate
too much sea ice (CCSM4 and UoT-CCSM4, orange), while other models simulate too little (CNRM-CM5, black; GISS-E2-
R, cyan; MIROC-ESM-P and MIROC-ES2L, purple; IPSL-CM5A-LR and IPSL-CM5A2, green). With only six proxy core
locations indicating SSI, it is not possible to extrapolate an estimate of the LGM sea-ice edge strictly based on this data.
However, we note that the highest individual model agreement is achieved by MRI-CGCM3 (blue), with an agreement of
98% and a sea-ice edge at 62.5˚S, which also corresponds to the PMIP3 MMM. On the other hand, the data-model agreement
significantly drops for models simulating a large sea-ice cover. CCSM4 (orange) and UoT-CCSM4 (orange), with sea-ice edges
of 55.5˚S and 53˚S respectively, simulate sea ice in locations where the proxy records suggest ice-free conditions in all sectors
of the SO. They are thus most likely over-estimating austral SSI cover.

### 3.3  Simulated LGM summer SO SST and comparison with proxy records

To better constrain the LGM SSI extent, we look into the proxy estimates of LGM summer SST data. We first assess the
relationship between zonally averaged simulated austral summer SSTs in the SO (between 50˚S and 75˚S) and the simulated
sea-ice edge and extent (Fig. 2a, b). The relationship between simulated summer SO SST and SSI edge or extent can be
approximated by a linear fit, with $R^2$ values of 0.90 and 0.81, respectively (Fig. 2a, b). Similarly, this relationship is also seen
during austral winter with $R^2$ values of 0.80 and 0.88, respectively (Fig. 5).

The mean LGM summer SO SST can be estimated from proxy records to be 1.52˚C $\pm$ 0.67˚C. Using the mean SST recon-
structed from the proxy records and the linear relationship estimated based on the simulations, we calculate a proxy SSI extent
estimate of 15.90 $10^6$ km$^2$ +/- 3.25 x $10^6$ km$^2$ and a mean sea-ice edge estimate of 61˚S +/- 2.25˚. The models closest to these
proxy sea-ice estimates are weakNA (15.73 x $10^6$ km$^2$ yellow plus sign, Fig. 2b), FGOALS-G2 (61.5˚S-pink triangle, Fig. 2a)
and AWI-ESM-1 (62˚S-dark pink square, Fig. 2a). LOVECLIM (yellow square) and weakNA_AB (yellow X mark) also fall
within the uncertainty of these estimates (Fig. 2a, b).

We also look into the meridional profiles of zonally averaged summer SSTs for all models (Fig. 2c, d). This is compared
to zonally averaged SSTs estimated from proxy data where SST proxy data is available (grey in Fig. 2c, d). The SST proxy
record suggests a mean SST of 1.44˚C south of 52˚S, with an increase of 1.1˚C per degree of latitude north of 52˚S. The SST
latitudinal variations in the models and proxies display significant differences, and none of the models included in this study
are able to reproduce the proxy distribution across all latitudes. Among the models, the distribution of zonally averaged SSTs is
not consistent over the SO. For example, at 75˚S both PMIP3 and PMIP4 models simulate a SST spread of around ∼3˚C while

at 65˚S both model groups simulate a SST spread of ∼6˚C (Fig. 2c, d). Between 52˚ and 65˚S, AWI-ESM-1 simulates a mean SST of 1.14˚C, closest to the proxy mean south of 52˚S. Some models are consistently warmer than the proxies (CNRM-CM5, black; GISS-E2-R, cyan; MIROC-ES2L, purple; IPSL-CM5A2, green), whereas others, with a large SSI cover, are colder than the proxy-based SSTs between 55˚S and 60˚S (CCSM4 and UoT-CCSM4, orange; CESM1.2, brown). Furthermore, most models are warmer than the proxies between 55˚S and 45˚S.

To look more closely into the regional distribution of SSTs, we plot each model's SO SST overlaid with the available SST data (Fig. 3). Due to biological limitations, diatom transfer functions are mostly available in regions with low sea-ice cover. As such, our proxy summer SST compilation only contains two locations with SST temperatures below 0˚C. With limited proxy SST data near the freezing point, we instead assess the model-data fit at the 1˚ isoline. We therefore compute the 1˚C isoline based on the proxies (solid black contour line in Fig. 3), and compare it to each model's 1˚C isoline (dotted black contour line in Fig. 3). The proxy data is regionally variable, with lower temperatures (darker blue points) in the Atlantic and Pacific sectors and higher temperatures (lighter blue, yellow, and red points) in the Indian sector. Additionally, there are more records at lower latitudes in the Atlantic and Indian sectors. Figure 3 confirms that certain models are too warm (CNRM-CM5; GISS-E2-R; MIROC-ESM-P and MIROC-ES2L; IPSL-CM5A-LR and IPSL-CM5A2) and certain models are too cold (CCSM4 and UoT-CCSM4; CESM1.2). The models that simulate a 1˚C isoline in good agreement with the proxy records are FGOALS-G2, AWI-ESM-1 and all LOVECLIM experiments (weakNA, weakNA_AB, LOVECLIM). To quantitatively establish how well the models simulate the SST proxy record, we calculate the root-mean-squared error (RMSE) between simulated SSTs and observations (Table 3). The model with the lowest RMSE value (1.40), representing the model for which simulated SSTs fit the reconstructions best, is the AWI-ESM-1 model. FGOALS-G2 also has a low RMSE value with 1.90.

Taking into account the spatial pattern of the sea-ice proxy record (Fig. 1), the regional variability of the SST proxy record (Fig. 2 and 3), and the RMSE scores from the SST proxy records (Table 3), the most likely LGM SSI edge lies at 61-62˚S, with a mean sea-ice extent of 14-15 x $10^6$ km$^2$, similar to the ones simulated by the AWI-ESM-1 or FGOALS-G2.

With such large SSI discrepancies among the models, we next look at the potential reasons for the observed inter-model spread. To identify drivers of inter-model sea-ice variability we analyze the thermodynamic and dynamic controls on sea-ice extent, as ocean temperatures exert a significant control on sea-ice formation and melt, and wind-stress affects sea-ice transport.

## 3.4 Drivers of inter-model variability

The strength and location of the southern hemispheric westerly and polar easterly winds impact SO circulation, sea-ice transport and therefore sea-ice distribution (Purich et al., 2016; Holland and Kwok, 2012). On the other hand, the presence or absence of sea ice also has a direct influence on surface winds (Kidston et al., 2011; Sime et al., 2016). Here, we focus on the influence of winds on sea ice through the divergence created by the wind stress curl. Within the SO, divergence leads to upwelling of relatively warm circumpolar deep waters and thus heat loss to the atmosphere. This upwelling can therefore also impact SO sea-ice distribution. While the latitudinal position and magnitude of southern hemispheric westerlies at the LGM is poorly constrained (Kohfeld et al., 2013; Sime et al., 2016), we want to assess the impact of the simulated windstress curl on ocean

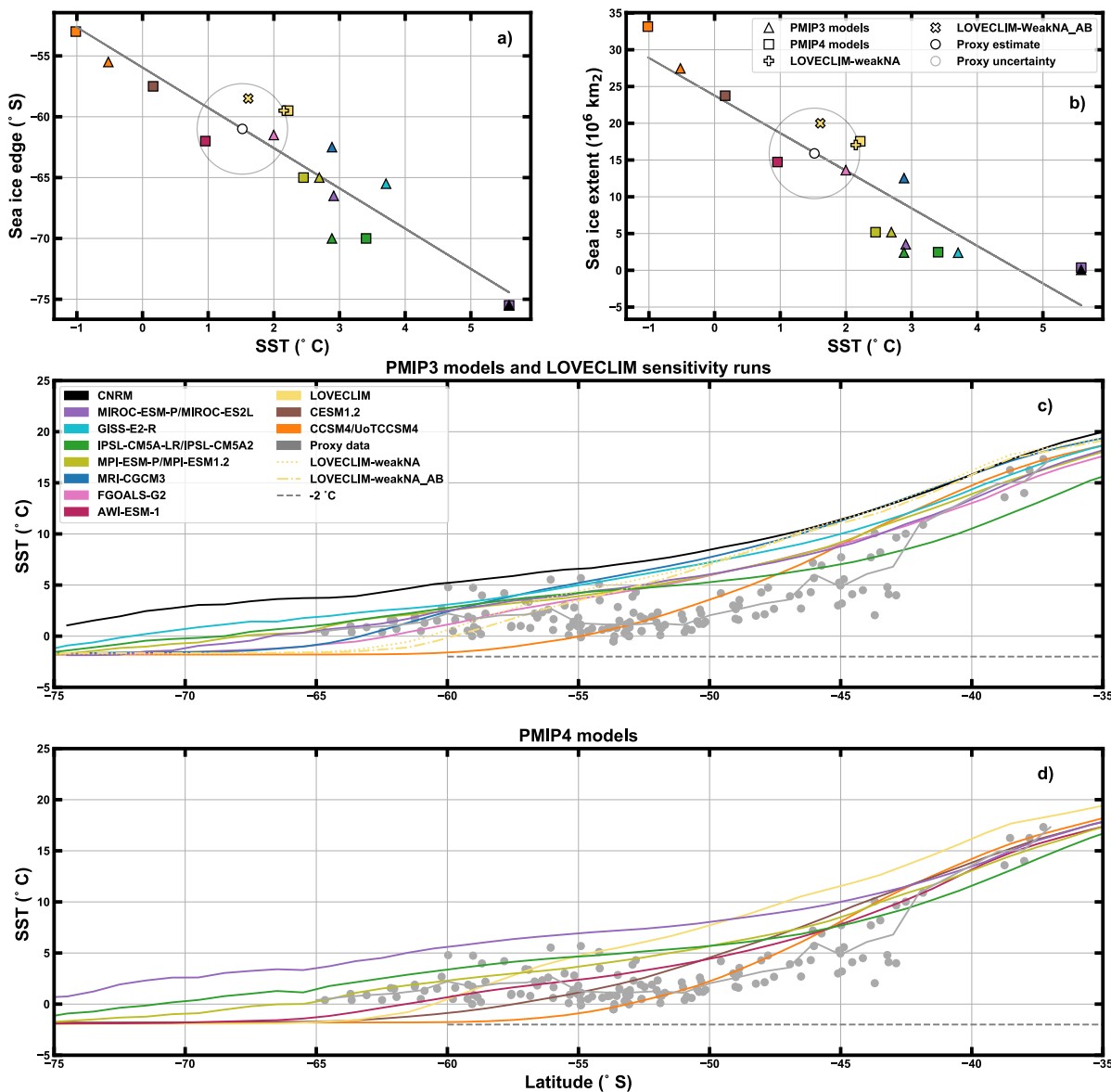

**Figure 2.** Austral summer sea ice and SST. a) Sea-ice edge vs. SST (50°-75°S); b) Sea-ice extent vs. SST (50°-75°S). Proxy summer sea-ice extent was estimated using the mean proxy SST value and the linear regression line. Uncertainty for the proxy SST value is shown in the grey circle. c) Zonally averaged SST values in the Southern Ocean as estimated from paleo-proxy records (grey circles) and for PMIP3 models and the LOVECLIM sensitivity experiments. The grey line represents the fit through the proxy data. d) Same as c) for PMIP4 models.

**Table 3.** Simulated seasonal sea-ice characteristics and austral summer SST, list from lowest to highest annual sea-ice extent. Model data agreement is calculated as the percentage of the correctly simulated sea-ice state at the location of proxies (presence of sea ice or ice-free conditions, with possible presence of sea ice considered ice-free). The austral summer Southern Ocean SST is meridionally averaged over 75˚S to 50˚S. The austral summer sea-ice edge is taken at the mean 15% concentration. Calculated root-mean-squared error (RMSE) values use the summer SSTs from proxy data in comparison to the modeled summer SST outputs.

| | Winter sea-ice edge (˚S) | Winter sea-ice extent ($10^6$ km$^2$) | Winter sea-ice agreement (%) | Summer sea-ice edge (˚S) | Summer sea-ice extent ($10^6$ km$^2$) | Summer sea-ice agreement (%) | Summer SO avg. SST (˚C) | Root mean square error |
|---|---|---|---|---|---|---|---|---|
| **PMIP3 models** | | | | | | | | |
| CNRM-CM5 | 53.5 | 23.01 | 65.77 | 75.5 | 0.06 | 95.45 | 5.58 | 4.03 |
| GISS-E2-R | 58.0 | 23.61 | 57.72 | 65.5* | 2.29 | 95.45 | 3.71 | 2.74 |
| IPSL-CM5A-LR | 52.5 | 27.62 | 78.52 | 70** | 2.41 | 95.45 | 2.88 | 2.23 |
| MIROC-ESM-P | 53.5 | 24.57 | 73.83 | 66.5** | 3.53 | 95.45 | 2.99 | 2.24 |
| MPI-ESM-P | 52.5 | 29.72 | 71.14 | 65 | 5.19 | 95.45 | 2.69 | 2.26 |
| MRI-CGCM3 | 50.0 | 36.50 | 84.56 | 62.5 | 12.54 | 97.73 | 2.89 | 3.53 |
| FGOALS-G2 | 50.5 | 32.86 | 86.58 | 61.5 | 13.62 | 93.94 | 2.00 | 1.9 |
| CCSM4 | 49.5 | 38.98 | 83.89 | 55.5 | 27.46 | 74.24 | -0.52 | 2.48 |
| **Multi-model mean** | 51.5 | 30.94 | 83.89 | 62.5 | 9.34 | 95.45 | 2.95 | 2.17 |
| **PMIP4 models** | | | | | | | | |
| MIROC-ES2L | 63.0 | 9.69 | 46.31 | 75.5 | 0.36 | 95.45 | 5.58 | 4.23 |
| IPSL-CM5A2 | 52.0 | 30.22 | 73.15 | 70.0** | 2.46 | 95.45 | 3.40 | 2.47 |
| MPI-ESM1.2 | 52.5 | 31.17 | 73.15 | 65.0 | 5.18 | 93.94 | 2.45 | 2.06 |
| AWI-ESM-1 | 51.0 | 34.57 | 82.55 | 62.0 | 14.73 | 95.45 | 0.96 | 1.40 |
| LOVECLIM | 52.5 | 32.18 | 79.19 | 59.5 | 17.55 | 90.91 | 2.22 | 3.67 |
| CESM1.2 | 50.0 | 38.47 | 85.23 | 57.5 | 23.75 | 78.79 | 0.16 | 2.10 |
| UoT-CCSM4 | 48.0 | 43.74 | 81.88 | 53.0 | 33.15 | 51.52 | -1.02 | 2.81 |
| **Multi-model mean** | 51.0 | 33.55 | 83.89 | 59 | 19.08 | 88.64 | 1.77 | 1.94 |
| **LOVECLIM sensitivity runs** | | | | | | | | |
| weakNA | 52.0 | 32.85 | 78.52 | 59.5 | 15.73 | 89.39 | 2.15 | 3.43 |
| weakNA_AB | 51.0 | 38.20 | 83.22 | 58.5 | 20.27 | 84.85 | 1.61 | 3.13 |
| **Multi-model mean** | 51.5 | 35.92 | 82.55 | 59 | 18.47 | 87.12 | 1.88 | 3.25 |
| **Proxy estimate** | - | - | - | 61.5 | 15.73 | - | 1.52 | - |

\* = Models with sea-ice edge calculated only in Ross Sea (150ºE to 220ºE)

\*\* = Models with sea-ice edge calculated only in Ross (150ºE to 220ºE) and Weddell Seas (290ºE to 360ºE)

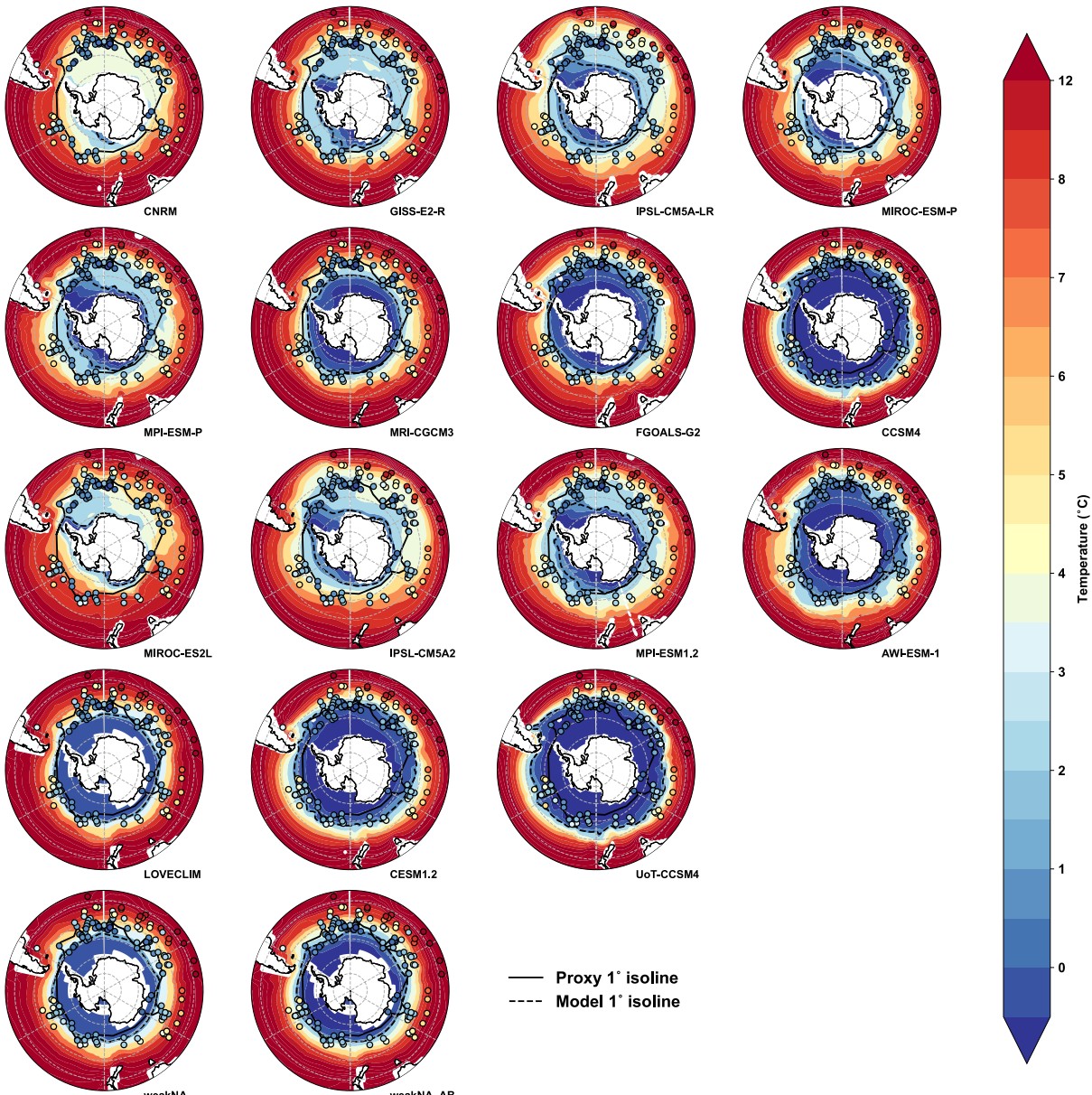

**Figure 3.** PMIP3, PMIP4 and LOVECLIM austral summer SST (shading) with SST proxy data overlain (filled circles). Fill color inside each SST reconstruction data point represents paleo SSTs at each location. Each model's 1˚C isoline (dotted black line) is compared to the proxy SST data 1˚C isoline (solid black line).

dynamics in each model. We thus use the windstress outputs to estimate the location and strength of the SO upwelling, and its potential impact on sea-ice cover.

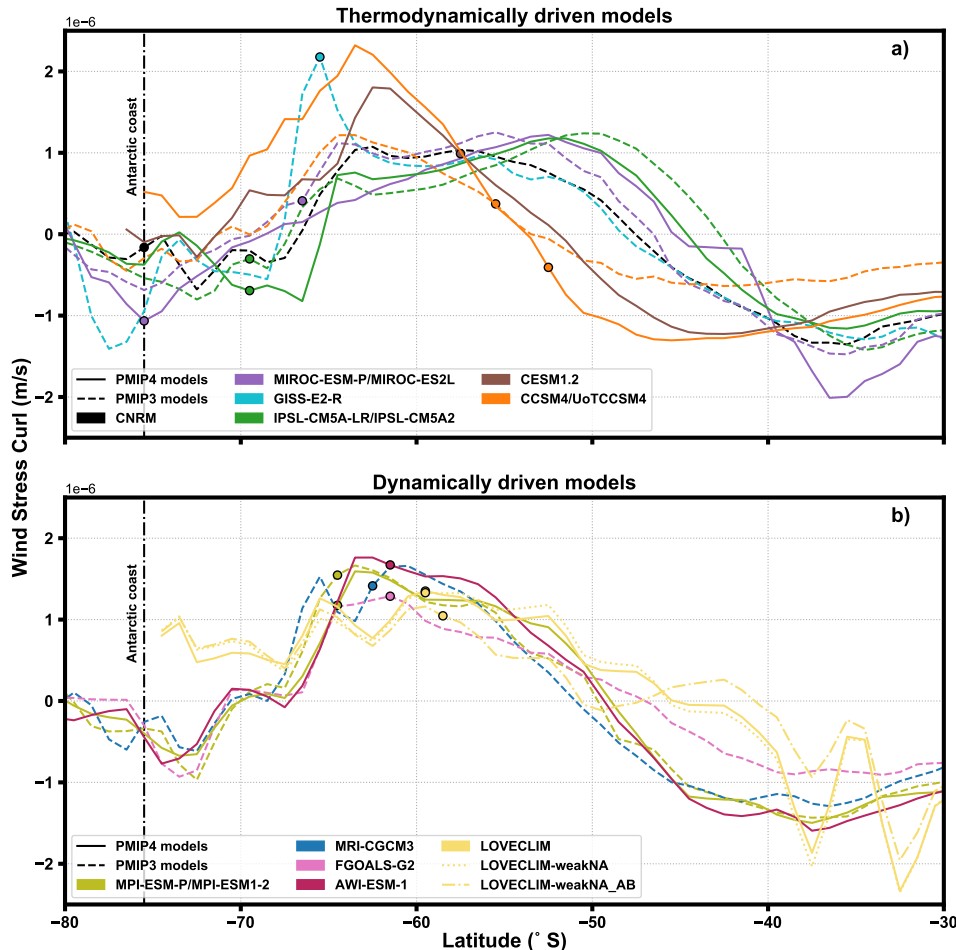

**Figure 4.** Zonally averaged austral summer wind stress curl vs. latitude for a) models in which mean temperature controls the summer sea-ice edge, b) models in which Southern Ocean upwelling, and associated divergence, impacts the summer sea-ice edge. Each model's sea-ice edge is represented with a filled circle.

Figure 4 shows the zonally averaged austral summer wind stress curl in the SO with each model's sea-ice edge overlaid. Figure 4b shows LGM experiments in which the SSI edge falls within 2-3 degrees of their zonal mean wind stress curl peak, indicating that sustained upwelling at 58°S - 65°S in these four models (MPI, FGOALS, AWI, LOVECLIM) most likely impacts summer SST and sea-ice cover. Our analysis suggests that the SSI edge in the MRI-CGCM3 LGM simulation is dynamically driven as its mean SO SST is close to the PMIP3 MMM, and its SSI edge is close to the maximum of the wind stress curl. However, this result should be taken with caution as in the MRI-CGCM3 simulation the coupling between sea ice and wind-stress at the ice/atmosphere interface was absent due to a model bug (Marzocchi and Jansen, 2017). On the other hand, Fig. 4a shows LGM experiments in which the SSI edge is more than 3 degrees away from the peak of the wind stress curl. The

models displayed in Fig. 4a both include LGM experiments with particularly high (CNRM-CM5, black; GISS-E2-R, cyan; MIROC-ESM-P and MIROC-ES2L, purple; IPSL-CM5A-LR and IPSL-CM5A2, green), and low (CCSM4 and UoT-CCSM4, orange; CESM1.2, brown) SST as identified in section 2. While the sea-ice edge in GISS-E2-R seems to occur at the maximum of the wind stress curl (cyan dotted line in Fig. 4a), this is an artefact of the averaging. This model's sea-ice edge is calculated only based on sea ice in the Ross Sea due to the lack of sea ice elsewhere. Therefore, the true global average sea-ice edge of GISS-E2-R at 15% concentration is at the Antarctic coast, which is more than 3 degrees from its wind stress curl peak. Divergence due to wind stress curl thus does not seem to have a large impact on SSI extent within the nine experiments (six models) shown in Fig. 4a.

We thus suggest that the location of the SSI edge in experiments displayed in Fig. 4a is thermodynamically controlled, whereas it is dynamically controlled for the ones displayed in Fig. 4b. It is interesting to note that experiments performed by the same models, or different versions of the same models, fall within the same categories: i.e. both PMIP3 and PMIP4 IPSL and MIROC experiments, as well as CCSM4 and UoTCCSM4 seem to be thermodynamically driven, whereas both PMIP3 and PMIP4 MPI and all LOVECLIM experiments seem to be dynamically driven.

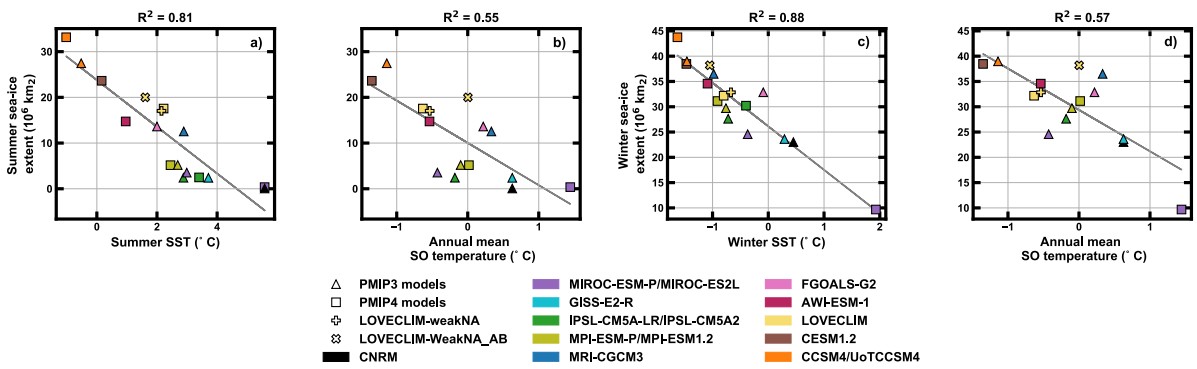

**Figure 5.** Scatter plot showing the relationship between sea ice, SO SST (averaged over 50-75˚S) and SO temperature (averaged over 50-75˚S and 0-5500m depth) in all experiments. Panels a, b show the relationship for austral summer and panels c, d show the relationship for austral winter.

As highlighted in Fig. 2, we find a clear relationship between seasonal sea-ice extent and seasonal SST. Additionally, as surface processes impact temperatures at deeper layers, we also find a statistically significant relationship between SO temperature (defined as the ocean temperature zonally averaged between 50˚S and 75˚S over the whole water column) and both WSI and SSI extent (Fig. 5). The larger the sea-ice extent, the lower the SO SST, and thus the lower the mean SO temperature. However, SO temperature is only partly controlled by AABW temperature but also by the latitudinal extent and temperature of circumpolar deep waters. For example, while they display very different sea-ice covers, both CCSM4 and GISS-E2-R simulate AABW close to freezing. However, in the GISS-E2-R simulation, the AABW extent is limited and the relatively warm NADW leads to warm circumpolar deep waters (Fig. 6).

The interplay between SO surface conditions and seasonal sea-ice cover modulates AABW temperature. LGM experiments with relatively cold surface conditions and relatively large SSI cover (e.g. CCSM4, LOVECLIM) also simulate cold ($< -1$°C) AABW. On the other hand, some LGM experiments with relatively warm conditions still simulate cold AABW due to the large seasonal difference in sea-ice cover (GISS-E2-R, MIROC-ESM-P). Finally, LGM experiments with both warm surface SO conditions and low seasonal differences in sea-ice cover simulate anomalously warm AABW (CNRM-CM5, MIROC-ES2L).

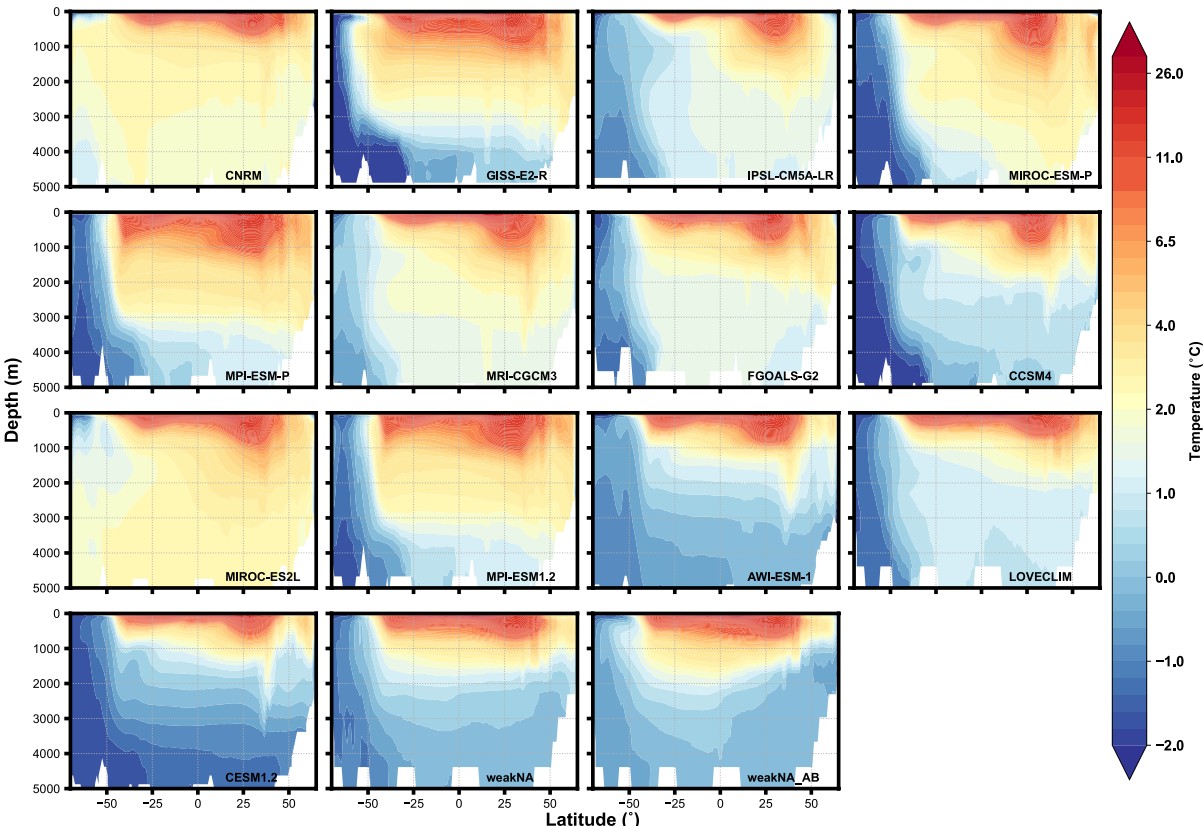

**Figure 6.** Zonally averaged oceanic potential temperatures (°C) in the Atlantic basin for PMIP3, PMIP4 and LOVECLIM simulations.

## 4   Discussion and conclusions

We suggest that during the LGM, the likely austral WSI edge was between 50.5°S and 51.5°S, with a mean sea-ice extent of 35-36 x $10^6$ km$^2$. During austral summer we suggest the sea-ice edge was likely between 61° and 62°S, with a mean sea-ice extent of 14-15 x $10^6$ km$^2$, similar to the sea-ice characteristics simulated by AWI-ESM-1 and FGOALS-G2. This is an improved constraint on LGM SSI extent as we combine modelling results with paleo-estimates of sea-ice cover and summer SST data. Previous LGM estimates, which were based only on sea-ice proxy data, are lower than ours (10.2-11.1 x $10^6$ km$^2$) (Lhardy

et al., 2021; Roche et al., 2012). Our estimates can also be compared to the average modern austral WSI extent of 18.5 x $10^6$ km$^2$ and the average modern austral SSI extent of 3.1 x $10^6$ km$^2$n (Eayrs et al., 2019).

Our estimate for the LGM SSI edge is a zonal average and therefore assumes a fairly circular SSI distribution, similar to that simulated by AWI-ESM-1 and FGOALS-G2 (Fig. 1). While the LGM SSI proxy data is limited, Lhardy et al. (2021) suggest the three basins behaved very differently, with a LGM SSI edge at 54˚S in the Atlantic, 65-66˚S in the Indian, 63˚S in the western Pacific and 66-68˚S in the eastern Pacific. If this indeed was the case, our suggested LGM SSI edge would potentially overestimate the sea-ice edge in some regions while potentially underestimating it in other regions. Additional proxy data from 320    the Pacific and Indian basins would reduce the uncertainty of our estimate.

    While the SSI edge is thermodynamically driven for six of the models considered here, it is linked to the position of the maximum windstress curl for the remaining five models (Fig. 5). The maximum wind stress curl corresponds to the maximum Ekman transport divergence, leading to deep-water upwelling. This can impact sea ice both thermodynamically and dynamically, as upwelling is often linked with ocean heat release, while the Ekman transport divergence can lead to strong equatorward 325    transport of sea ice. Given the uncertainties that surround the magnitude and position of the Southern Hemisphere westerlies at the LGM (e.g., Kohfeld et al., 2013; Sime et al., 2016), this casts additional uncertainties on the location of the austral SSI edge. Furthermore, paleo records of austral SSI extent used here are mostly restricted to 40˚- 60˚S, with 95% of the records suggesting ice-free conditions. Due to this, they can only provide an estimate of the maximum SSI extent. Additional proxy records recovered from locations south of 60˚S are thus needed to better constrain the SSI extent.

Both the PMIP4 and PMIP3 experiments display a relatively large range ($\sim$ -2˚ to 4˚C) of temperatures in the deep Atlantic Ocean (Fig. 6). Only a few paleo-records of deep ocean temperature are available for the LGM, but they suggest ocean temperatures below 0˚C throughout the deep Atlantic (Adkins et al., 2002). In the southwest Pacific at ODP Site 1123, Mg/Ca records find deep ocean temperatures of -1.1 $\pm$ 0.3˚C at the LGM (Elderfield et al., 2010). The models that simulate warm SO conditions with little sea ice also simulate a warm bias at depth, even though in some cases a large seasonal ($\sim$20 x $10^6$ km$^2$) 335    difference between maximum and minimum sea-ice extent can lead to cooler abyssal temperatures.

    In this study, we also included three LGM experiments performed with the Earth system model LOVECLIM. The oceanic circulation was varied in two of these experiments by adding meltwater in the North Atlantic and SO and weakening the southern hemispheric westerly windstress (Menviel et al., 2017). Despite significant differences in oceanic circulation in these three simulations, with weaker AABW transport in weakNA_AB compared to weakNA, and weaker Atlantic meridional overturn- 340    ing circulation (AMOC) in weakNA (14.7 Sv) and weakNA_AB (11.2 Sv) compared to the PMIP4 LOVECLIM experiment (26 Sv), the differences in sea-ice extent between these three experiments are much smaller than the inter-model differences between all PMIP3 and PMIP4 simulations. This indicates the limitations of performing model-data comparisons with a single model to infer SO climatic conditions.

    We further assess the relationship between LGM SSI and AMOC strength (Fig. S2, Muglia and Schmittner 2015; Kageyama 345    et al. 2021), and find that there is no statistically significant relationship between the two (R$^2$=0.04). There is however a weak relationship between SSI extent and AMOC depth (Fig. S2, R$^2$=0.17), with a shallower AMOC generally associated with a larger SSI extent. A larger SSI extent, and thus increased sea-ice formation, could impact the AABW properties and therefore

ocean stratification (Marzocchi and Jansen, 2017), as evident from Fig. 5. However, climatic conditions in the North Atlantic are probably the principal driver of AMOC depth (Oka et al., 2012; Muglia and Schmittner, 2015). There is also no link between the equilibrium climate sensitivity (ECS) of these models and their austral SSI cover, with the three models displaying the least amount of sea ice exhibiting ECS of 3.3˚C for CNRM-CM5, 2.7˚C for MIROC-ES2L, and 2.1˚C for GISS-E2-R, while the two models with the most sea ice have an ECS of 2.9˚C and 3.2 (CCSM4 and UoT-CCSM4, respectively, Kageyama et al. 2021).

Based on our best estimates of LGM WSI and SSI cover over the SO, the seasonal variation in sea-ice edge is $\sim$10˚ and the seasonal variation in sea-ice extent is 20-22 x $10^6$ km$^2$. In comparison, the present day seasonal change in sea-ice edge ranges from $\sim$15˚ in the Atlantic sector to less than 5˚ in the Indian sector (Cavalieri and Parkinson, 2012). The present day seasonal variation in sea-ice extent is $\sim$15.4 x$10^6$ km$^2$ (Eayrs et al., 2019), thus indicating a larger sea-ice seasonality during the LGM. Such a large sea-ice seasonality would in turn impact SO dynamics through changes in buoyancy (Marzocchi and Jansen, 2017) as well as the carbon cycle (Haumann et al., 2016). While a large year-round LGM sea-ice cover could contribute to a lower atmospheric $CO_2$ concentration (Ferrari et al., 2014), the impact of a large sea-ice seasonality on the carbon cycle is not well constrained. The increased seasonality has potential to dampen $CO_2$ drawdown, depending on the balance between up-welling and subsequent outgassing of carbon-rich deep waters and nutrient utilization at the surface (e.g., Menviel et al., 2008). Conversely, the increased seasonality could also amplify carbon drawdown through enhanced brine formation, increasing the density gradient between the surface and deep waters (Galbraith and de Lavergne, 2019), and potentially lowering atmospheric $CO_2$ (Bouttes et al., 2012). Despite proxy records showing lower productivity in the Antarctic Zone (Jaccard et al., 2013), increased stratification due to sea-ice melt during spring-summer could enhance nutrient utilization and thus carbon drawdown (Sigman and Boyle, 2000; Abelmann et al., 2015). While some studies have suggested a primary role for LGM sea-ice cover in driving changes in oceanic carbon content (Ferrari et al., 2014), the LOVECLIM experiments presented here have also shown that reduced ventilation of the deep ocean through weaker AABW transport could instead be the primary driver of an increase in deep ocean carbon content (Menviel et al., 2017).

Antarctic sea ice integrates oceanic and atmospheric processes occurring at high southern latitudes, and can also significantly impact Antarctic climate (Bracegirdle et al., 2015). Sea ice has the ability to protect ice-shelves (Massom et al., 2018) and floating ice-shelves play a significant role in buttressing Antarctic outlet glaciers (Scambos et al., 2004). It is thus crucial that models incorporate a good representation of pre-industrial and present-day sea ice, but also manage to correctly simulate past sea-ice extent during both cold periods, such as the LGM, and warm periods such as the Last Interglacial (125,000 years ago). In that regard, it is interesting to note that the models which underestimate austral summer Antarctic sea-ice cover at the LGM also underestimate the austral SSI cover under pre-industrial conditions, while the model simulating the largest LGM sea-ice cover also overestimates the pre-industrial SSI cover (Marzocchi and Jansen, 2017; Goosse et al., 2013; Roche et al., 2012). This implies that targeting a good agreement between model and observations for present day climate should remain a priority.

In this study, we have analysed SO WSI and SSI cover in LGM simulations and compared the outputs against available proxy reconstructions. In doing so, we identify the potential drivers for inter-model SO sea-ice differences, in addition to placing improved constraints on the LGM SO SSI and WSI extents. This improved understanding of sea-ice dynamics can

provide valuable information about the Earth's system and important insight into the strengths and weaknesses of models currently used.

*Data availability.* The ocean and sea-ice data for the LOVECLIM sensitivity runs can be found at https://doi.org/10.26190/K6XA-T076. The PMIP3, PMIP4 and LOVECLIM multi-model mean data can be found at https://doi.org/10.26190/unsworks/1636. PMIP3 data can be found at https://esgf-node.llnl.gov/search/cmip5/ and most PMIP4 data can be found at https://esgf-data.dkrz.de/search/cmip6-dkrz/.

*Author contributions.* RAG performed the data analysis. LM and KJM conceived the study and provided support to the interpretation of results. XC compiled existing sea-ice proxy data and provided expert knowledge on sea-ice processes and sea-ice proxy data. RAG wrote 390 the manuscript with contributions from LM, KJM and XC. DC, GL, WRP, XS and JZ provided the PMIP4 model outputs presented in this paper. All authors contributed to the final version of the manuscript.

*Competing interests.* The authors declare that they have no conflict of interest.

*Acknowledgements.* This research is a result of the Past Global Changes (PAGES) working group 'Cycles of Sea-Ice Dynamics in the Earth system' (C-SIDE). We thank Masa Kageyama for providing the PMIP4 LGM outputs of the IPSL model. Ryan Green was supported by a 395 summer scholarship provided by the Australian Research Council Centre of Excellence for Climate Extremes (CE170100023), and UNSW (through Laurie Menviel's UNSW Scientia fellowship). Laurie Menviel and Katrin Meissner are thankful for funding from the Australian Research Council (FT180100606, DP18010004, DP180102357). Computational resources were provided by the NCI National Facility at the Australian National University, through awards under the National Computational Merit Allocation Scheme, the Intersect allocation scheme, and the UNSW HPC at NCI Scheme.

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
