# Peer review of "Evaluating seasonal sea-ice cover over the Southern Ocean at the Last Glacial Maximum"

_Climate of the Past, 2020_

## Short Comment (SC1) · 16 Jan 2021

In the Discussion section the authors state that "Despite significant differences in oceanic circulation in these two simulations, with weaker NADW and AABW in LOVE-CLIM2 compared to LOVECLIM1, the sea-ice cover differences between these two runs are much smaller than compared to other models. Apart from FGOALS-G2, which simulate a very strong LGM AMOC, the LGM AMOC strengths in the other PMIP3 models are similar at 21-23 Sv" Although the two LOVECLIM simulations have different AABW circulations, they both have weak AMOCs (taken from Menviel et al. 2017 table 1). So it is natural that the difference in sea ice cover between these two simulations is smaller than between them and the PMIP3 models, which all have stronger AMOCs. Could you please be more clear on how this result means that the "primary control on

LGM austral summer sea-ice cover is not linked to the strength of the AMOC"? Especially taking into account that from Fig. 1c it looks like the two simulations that have the furthest reaching Summer sea ice are LOVECLIM1, LOVECLIM2 (both have weak and shallow AMOC), and CCSM4 (strong and shallow AMOC), which are the three simulations with the most distinct AMOC structures compared with the rest of the PMIP3 models which exhibit strong and deep AMOCs.

Thanks!

---

## Referee Comment (RC1) · Anonymous Referee #1 · 5 Feb 2021

Green et al presents a new valuable compilation of summer and winter sea ice and SST data for the Southern ocean during the LGM. The paper is, for the most part, quite well written and presented. I very much like that both sea ice and SST data are compiled and then used to evaluate model simulations. The manuscript is however, in its current form, sometimes rather weak on explaining its aims. I do nevertheless find it is suitable in principle for Climate of the Past, after some revision.

I have three main comments on this paper.

(1) There is no suitably clear explanation of what is new, or what is being evaluated, or indeed why, in the introduction or abstract. Clearer sign-posting about the paper containing a new compilation of sea ice and SST data for the LGM would also be helpful in the abstract. I found it a difficult to tell whether the new marine data were

being evaluated for internal consistency. Or whether CMIP5-PMIP3models were being evaluated for their ability to simulate Summer and Winter across the Southern Ocean sea ice in the LGM. Or perhaps whether the ability of models to simulate a large (or small) seasonal change in the area or extent of sea ice in the Southern Ocean was being tested. Its also unclear what the motivation is for evaluating models.

(2) Closely linked to (1), the choice of models being evaluated is difficult to defend. Evaluating CMIP5-PMIP3 versus CMIP6-PMIP4 – and whether models have improved in their ability to simulation SO sea ice would seem valuable and more easy to understand as motivation. However the paper in its current form does not do this. Instead, it shows results from the older PMIP3 simulations, alongside some more recent LoveClim simulations. At least one of the LoveClim simulations also has a very strange seasonal cycle of sea ice. The motivation for doing this approach, as opposed to PMIP3 versus PMIP4 is never explained. I would encourage the authors to consider focussing on PMIP4 versus PMIP3/2. More than 10 PMIP4 LGM simulations are available to the authors.

(3) The section on wind stress results is not particularly helpful. It needs more on how glacial-interglacial wind changes depend on model biases, and the sea ice itself. It can help to start with considered wind velocities above the surface, before then considering wind stress/curl changes. Otherwise results can tend to confuse.

Minor comments

L90-~95. Be specific about what 'common' (e.g. 1 in 2 years?) and 'episodic' (e.g. 1 in 10 years, 1 in 50 years?) mean.

Table 2 – It is most strange that LOVECLIM1 has a SIE minimum in Jan-Feb. Suggest excluding this model from the analysis. Hard to see how the results can be meaningful.

L110 "The proxy"

Figure 1 – add latitude grid lines.

L160 onward. Given the freezing point of seawater is ∼-1.8C, I find the lack of <0C datapoints a little strange. Would be useful to have some discussion of the relationship between the freezing point (∼-1.8C) and the model and obs datapoints here.

Figure 2. Consider adding a -1.8C or -2C line on panel (c).

Figure 2. Obs data is too faint on panel (c). Make this more visible please.

Table 3 – order of columns seems strange. Maybe should be Winter: SI edge, SIE, SIA, Obs agreement, SST, SST agreement; then Summer: SI edge, SIE, SIA, Obs agreement, SST, SST agreement.

Figure 3 – Cannot see Southern Ocean temperature variation with current colorbar. Change scale to -2 to +8? Figure not useful with this colorscale.

Line 229-253 There are three main problems with this section. Firstly it currently lacks adequate discussion of the fact that SHWs are too poorly simulated by most PMIP3 models to be able to straightforwardly interpret wind results. Secondly, the discussion needs to take account of the control that sea ice exerts on SH winds. Thirdly, by focussing on stress and curl, rather than simpler above surface speed/velocity measures (which are less affected by surface roughness) it leads to a discussion that probably confuses the reader more than clarifying anything.

Suggest re-reading relevant SO/sea ice wind papers by Bracegirdle et al (2013, 2018 and others), Sime et al (2013, 2016), and Kidston et al (2011, The influence of Southern Hemisphere sea-ice extent on the latitude of the mid-latitude jet stream, Geophys. Res. Lett., 38, L15804, doi:10.1029/2011GL048056).

Rewrite or remove this section?

---

## Referee Comment (RC2) · Anonymous Referee #2 · 10 Feb 2021

General Comments

This study compares PMIP3-simulated sea ice cover in the Southern Ocean, as well as
that from two LOVECLIM experiments, against an updated catalogue of sea ice paleo-
proxy data. In that sense, it follows directly on Roche et al (2012) and Marzocchi and
Jansen (2017). They focus on the summer season, as that season has had the fewest
data constraints in previous studies. Additionally, they correlate the sea ice edge to
SST and wind stress curl in the models.

Overall, I find the results of this study underwhelming. Although they provide a lot
of results, I feel that not a lot of meaning is derived from them (i.e. Too much of a
"figure tour" or "data tour"). Assessing the paleo-proxy data is outside my area of
expertise, but the model analysis does not feel like a substantial contribution beyond

Marzocchi and Jansen (2017), who examined Southern Hemisphere sea ice controls in these same PMIP3 simulations. Indeed, I find it concerning that a serious flaw in one of those simulations reported in Marzocchi and Jansen (2017) (that of a bug-related absence of wind-stress feedback on sea ice in MRI-CGCM3) is not mentioned here, even though wind-stress curl is one of the foci of the study. The two additional LOVECLIM simulations do not appear to be used in any significant way to get at causal mechanisms, even though these simulations differ substantially in both the design of the experiments and complexity of the model.

Finally, the authors do not provide much discussion of their results in the context of the previously-mentioned two papers, and given their claim that "the multi-model mean of austral summer and winter sea ice cover seem to provide good estimates of LGM conditions" appears to be at odds with the results presented in those papers for earlier generation (Roche et al, 2012) and the same model (Marzocchi and Jansen, 2017) data, I would have liked a bigger exploration of this difference. I found it hard to interpret these differences on my own due to the small size and indistinguishable lines in the figures and vague descriptions of methodologies.

Specific Comments

intro raises connections w/ SAM – why not connect results to SAM?

Line 55: while Marzocchi and Jansen didn't focus on seasonality of PMIP3 simulations, they did plot the seasonal comparisons btw PI and LGM and discuss the characteristics of the seasonality in the models

Given zonal asymmetries in sea ice edge, usefulness of hemispheric, zonal averages is unclear

Line 67-69: Multiple simulations from same model were averaged over – what if they involved different components? How reflective is the ensemble mean of the behaviour of any one simulation? (look into GISS-E2-R)

The LOVECLIM simulations that were chosen are not representative of the PMIP3 models for a number of reasons, and the differences in their attributes, the reasons for their selection and the implications for the results are not discussed much here. As a result, the comparison feels artificial and not very instructive. Looking further into the simulations in Menviel et al (2017), I can make a guess as to why these ones were chosen, on the basis of the performance of their carbon cycle models. However, the fact that these simulations were performed with prognostic $CO_2$ concentrations (via a carbon cycle model ) rather than prescribed emissions/concentrations, and were spun up in a transient fashion from 35ka BP rather than just equilibrated to fixed LGM conditions as most of the PMIP3 simulations would have been, and had anomalous hosing applied in either or both of the Northern and Southern Hemispheres has bearing on the interpretation of the results. However, only the hosing is mentioned, and the implications of these design choices on the sea ice distributions are not discussed.

I am concerned about the fact that at least one lower-resolution model (e.g. LOVE-CLIM, whose ocean is nominally 3degx3deg) was interpolated onto a higher-resolution grid (1degx1deg) for plotting the results. This artificially inflates the apparent resolution of the results and thus encourages attempts to interpret changes at smaller spatial scales than the model provides as real. Whether the resolution of any PMIP3 models was artificially inflated in this way is not clear.

Lines 82-97 I am not a specialist in the interpretation of sea ice proxy records, so I found this section confusing. My main source of confusion lay in interpreting the uncertainties related to the apparently weak signals (differences between 1-3% in diatom assemblages), given the RMSEP values were 10%. I'm assuming the two percentages were not referring to the same quantities. I'm not expecting this paper to provide an overview of this method, assuming the references already provide that, but a sentence or two to make these results interpretable to non-specialists would be appreciated.

Why were two months selected to define sea ice maxima and minima? Was this based on prior knowledge, or analyses of the seasonal cycle in the models?

Lines 104-106 I'm not entirely clear on the methodology here. Firstly, were zonal averages performed in each ocean basin region over the latitudes of 15% sea ice concentrations for each longitude division (= 1 grid cell) or over sea ice concentrations in each latitude band (from which the 15% was then calculated)? And then, when defining an hemispheric sea ice latitude, am I correct in understanding that a zonal average over all longitudes was not calculated and instead, an average (weighted or unweighted?) was performed over the individual ocean basin regions? If this is correct, why was this method chosen?

Lines 118-119 Why does the multi-model mean lead to an asymmetric region of variance north and south of the mean lines? Are you suggesting that the simulations are distributed in a non-Gaussian way? Based on visual inspection, I wonder how the mean was calculated. I'm assuming the multi-model mean was calculated by averaging over the latitude of sea ice margin for each cell's longitude range from each run, but it doesn't seem to match what I see in the figure. For example, between 150 and 180degE, two, maybe 3, runs extend past the latitude of the outermost blue points. That leaves 6ish runs south of those points, but the multi-model mean lies north of the points. If the two runs were far north of the points, I would understand this, but that is not the case. Rereading the text, I wonder if the authors are suggesting they performed a multi-model average of sea ice concentration in every grid cell and then calculated the 15% concentration margin from the ensemble-mean distribution. If those more extensive runs had very high sea ice concentration up until their margins, it might explain the position of the mean, but it's not clear to me how the standard deviations of the 15% line could be derived from this calculation. Since assigning a cutoff to sea ice at the 15% concentration is a non-linear operation, I would expect to get a more Gaussian multi-model distribution for the latitudes of the 15% lines than performing the calculation on the ensemble means of the sea ice concentrations directly.

Given the zonal asymmetries in the summer data and their relative absence in the models (based on inspection of the multi-model mean in Figure 1 and the performance

of PMIP2 models in Roche et al, 2012), I'd like to see the analysis performed for Figure 2 calculated on a regional basis, rather than based on hemispheric averages. Such an analysis would be more likely to bring out discrepancies between the models and the data than the hemispheric average.

Technical Comments

Two different definitions for LGM time period provided in the abstract and text

I find it very difficult to distinguish between the colours of the different lines in Figure 1, because they are so thin. If thickening was not chosen because of confusion in the plot, I think the line labels can easily be dropped to simplify the figure. Also, without any latitudes and longitudes marked on the plot and only the southern tip of S. America included as a georeference, it takes a lot of work for someone who is not intimately familiar with Antarctic geography to translate the descriptions in the text (most of which seemed to be in longitudes) to their locations on the figure and compare the latitude values stated in the text with those in the figure.

It would be helpful for the boundaries of the ocean basin regions to be marked in one figure. Their names are clear, but precisely where the boundaries between the basins are drawn and their northern and southern extents would be helpful in interpreting the results.

Line 110: Sentence fragment "The proxy"

Line 214: Typo MRI-ESM-P should be MRI-CGCM3

---

## Referee Comment (RC3) · Anonymous Referee #3 · 12 Apr 2021

Green et al., propose some interesting insights related to seasonal changes in sea-ice cover during the LGM using PMIP3 and LOVECLIM simulations, in particular in relation to wind stress and surface ocean temperatures. While simulations are generally coherent with paleo reconstructions for LGM winter sea-ice cover, simulated glacial summer sea-ice cover differs widely between models and between models and (arguably limited) proxy-based reconstructions.

While study is certainly relevant, it lacks some context, I feel. To me, the main outcome of the present study relates to the conclusion that the seasonal contrast in sea-ice cover is reduced today compared to the LGM. While clearly interesting, this finding should be placed into a broader context and outline how increased sea-ice dynamics

would affect ocean circulation and more generally carbon and nutrient biogeochemistry during the LGM. Furthermore, it remains unclear why reconstructing summer sea-ice extent is relevant, in particular for air-sea gas exchange and deep water production, which to a large degree occur in winter.

I certainly support publication of the present study in Climate of the Past, provided some additional contextualization can be provided.

General comment

I'm certainly not a modeling expert, but the reason underlying the poor representation of the glacial summer sea-ice extent should be clarified. In particular, I don't quite understand why the distribution of mean summer SST during the LGM is generally coherent between models while summer sea-ice extent is not. Maybe this aspect could be clarified further at least for non-specialized readers.

Detailed comments

l. 15-27. Since the MS is focusing on seasonal contrasts in sea-ice extent, it may be relevant to briefly explain how seasonality affects the main processes outlined in this section today. For example, does a positive SAM phase affect both the winter and summer sea-ice extent linearly?

l. 27-30. The statement is misleading. I believe the Southern Ocean has accounted for about 40% of the global OCEANIC uptake of anthropogenic $CO_2$.

l. 44. Could you briefly explain why records of summer sea-ice extent are generally more poorly constrained?

l. 110 – incomplete sentence

l. 161-163 & 168-169 – how does the relationship between SST and sea-ice cover equate for the winter months as a comparison?

l. 203 – how does this value compare with modern SSI extent?

l. 321-323 – assuming that most of the mixing occurs during the winter, I'm not too sure to understand how increased sea-ice melt during the spring-summer could enhance nutrient utilization (and by inference carbon drawdown)?

————————————————————

**[CPD](CPD)**

---

## Author Comment (AC1) · 30 Nov 2021

We thank Dr. Juan Muglia for his comment. His comments are reproduced in black, and our replies are provided in blue. We have also included the modified section of the revised manuscript below our response.

**Comment SC1: Juan Muglia (16 January 2021)**

In the Discussion section the authors state that "Despite significant differences in oceanic circulation in these two simulations, with weaker NADW and AABW in LOVECLIM2 compared to LOVECLIM1, the sea-ice cover differences between these two runs are much smaller than compared to other models. Apart from FGOALS-G2, which simulate a very strong LGM AMOC, the LGM AMOC strengths in the other PMIP3 models are similar at 21-23 Sv" Although the two LOVECLIM simulations have different AABW circulations, they both have weak AMOCs (taken from Menviel et al. 2017 table 1). So it is natural that the difference in sea ice cover between these two simulations is smaller than between them and the PMIP3 models, which all have stronger AMOCs. Could you please be more clear on how this result means that the "primary control on LGM austral summer sea-ice cover is not linked to the strength of the AMOC"? Especially taking into account that from Fig. 1c it looks like the two simulations that have the furthest reaching Summer sea ice are LOVECLIM1, LOVECLIM2 (both have weak and shallow AMOC), and CCSM4 (strong and shallow AMOC), which are the three simulations with the most distinct AMOC structures compared with the rest of the PMIP3 models which exhibit strong and deep AMOCs.

We thank Juan Muglia for this comment. Since most PMIP3 models (apart from FGOALS) have similar AMOC strengths but very different sea-ice areas (Fig. R1), other factors than AMOC must control the summer sea-ice extent. For the two LOVECLIM simulations originally included, we were mostly referring to AABW, which is much weaker in weakNA_AB (previously referred to as LOVECLIM2) than weakNA (previously referred to as LOVECLIM1). Nevertheless, we are now also including another LOVECLIM experiment following the PMIP4 protocol, which displays an AMOC strength of 26 Sv (compared to 14.7 Sv and 11.2 Sv for weakNA and weakNA_AB) to illustrate the fact that despite the very different AMOC states, differences in sea-ice extent simulated in these 3 LOVECLIM simulations are smaller than across models.

As shown in Figure R1 (below, Figure S1 in the new version of the manuscript), we are now also plotting the sea-ice extent as a function of AMOC strength and depth for all the simulations. Figure R1 shows that across PMIP3 and PMIP4 LGM simulations, there is no relationship between sea-ice extent and AMOC strength, and only a weak relationship between sea-ice extent and AMOC depth.

In the revised manuscript, the Discussion will be amended as follows:

"In this study, we also included three LGM experiments performed with the Earth system model LOVECLIM. The oceanic circulation was varied in two of these experiments by adding meltwater in the North Atlantic and Southern Ocean and weakening the southern hemispheric westerly windstress (Menviel et al., 2017). Despite significant differences in oceanic circulation in these three simulations, with weaker AABW transport in weakNA_AB compared to weakNA, and weaker AMOC in weakNA (14.7 Sv) and weakNA_AB (11.2 Sv) compared to the PMIP4 LOVECLIM (26 Sv), the differences in

sea-ice extent between these three experiments are much smaller than the intra-model differences between all PMIP3 and PMIP4 simulations. This indicates the limitations of performing model-data comparisons with a single model to infer SO climatic conditions."

The lack of relationship between sea-ice extent and AMOC strength, and the weak relationship with AMOC depth are now also mentioned in the Discussion and shown in Figure S4:

"We further assess the relationship between SO sea-ice extent and AMOC strength (Figure S4, Muglia and Schmittner, 2015; Kageyama et al., 2021), and find that there is no statistically significant relationship between the two ($R2=0.04$). There is however a weak relationship between sea-ice extent and AMOC depth (Fig. S2, $R2=0.17$), with a shallower AMOC generally associated with a larger sea-ice extent. SO sea-ice formation impacts AABW properties and therefore ocean stratification (Marzocchi and Jansen 2017), which can influence AMOC depth. To some extent, this can be seen in Figure 5, as models that simulate small amounts of sea ice (i.e. CNRM and MIROC-ES2L) show less stratification and a deeper AMOC, while models simulating more sea ice (i.e CCSM4 and CESM1.2) have more stratification and a shallower AMOC. However, climatic conditions in the North Atlantic are probably the principal driver of AMOC depth (Oka et al., 2012; Muglia and Schmittner, 2015).

[Figure]

Figure R1 (shown in the manuscript as Figure S4). AMOC strength (Sv) and depth (m) at 25˚N vs. summer sea-ice extent in the Southern Ocean ($10^6$ km²)

---

## Author Comment (AC2) · 30 Nov 2021

We thank the reviewer for these useful comments which will certainly improve the manuscript. The reviewers' comments are reproduced in black, and our replies are provided in blue. When necessary, we have also included the modified section of the revised manuscript below our response.

**Review RC1 (5 February 2021)**

Green et al. presents a new valuable compilation of summer and winter sea ice and SST data for the Southern ocean during the LGM. The paper is, for the most part, quite well written and presented. I very much like that both sea ice and SST data are compiled and then used to evaluate model simulations. The manuscript is however, in its current form, sometimes rather weak on explaining its aims. I do nevertheless find it is suitable in principle for Climate of the Past, after some revision.

Thank you for these comments. In the revised version of the manuscript, we are providing more context and making the manuscript's goals more explicit within the abstract, introduction, and throughout the results.

I have three main comments on this paper.
(1) There is no suitably clear explanation of what is new, or what is being evaluated, or indeed why, in the introduction or abstract. Clearer sign-posting about the paper containing a new compilation of sea ice and SST data for the LGM would also be helpful in the abstract. I found it a difficult to tell whether the new marine data were being evaluated for internal consistency.

We thank the reviewer for this helpful comment. In the new version of the manuscript, we now explicitly state what is new in our analysis. Please refer to our answers below for more detail.

Regarding the internal consistency of the proxy data, the data presented builds on the previous compilation of Gersonde et al. (2005). This compilation was expanded by adding recently published data (Allen et al. (2011), Ferry et al. (2015), Benz et al. (2016), Xiao et al. (2016), Nair et al. (2019), and Ghadi et al. (2020)). This data has been re-evaluated during this work for (1) their chronological framework, (2) their methodology, and (3) their consistency with surrounding data. The studies are based on very similar methodologies ensuring a strong homogeneity.

This was clarified in the Introduction:
"The LGM sea-ice compilation covering the Southern Ocean of Gersonde et al. (2005) is routinely used to provide estimates of LGM sea-ice cover. New Southern Ocean sea-ice data, reconstructed using statistical methods and diatom assemblages, has however been published since then (Allen et al., 2011; Benz et al., 2016; Ferry et al., 2015; Xiao et al., 2016; Ghadi et al., 2020; Nair et al., 2019) thus highlighting the need for an updated compilation."

Or whether CMIP5-PMIP3 models were being evaluated for their ability to simulate Summer and Winter across the Southern Ocean sea ice in the LGM. Or perhaps whether the ability of models to simulate a large (or small) seasonal change in the area or extent of sea ice in the Southern Ocean was being tested. Its also unclear what the motivation is for evaluating models.

We appreciate the reviewer pointing out this confusion. To clarify, our manuscript has several goals: 1) to provide a spatial and seasonal assessment of LGM Southern Ocean sea-ice extent in PMIP3 models (and now also PMIP4 models), 2) to compare available sea-ice and SST records to simulations to assess the skill of each model and of the ensemble, 3) to understand the processes leading to the large range in summer sea-ice extent amongst different model simulations.

This is now clarified in the Introduction:

"Here, we assess the minimum and maximum Southern Ocean sea-ice extent as simulated in LGM PMIP3 and PMIP4 experiments. To better assess intra- versus inter-model variability, LGM sensitivity experiments performed with LOVECLIM are also included. The PMIP3, PMIP4 and LOVECLIM experiments are compared to available sea-ice and SST paleo-proxy data, allowing us to determine the best model-data fit. Combining model and proxy data, we can provide an updated estimate of seasonal Southern Ocean sea-ice cover during the LGM. Furthermore, we analyze the processes that lead to the inter-model differences in summer sea-ice extent at the LGM."

Regarding the motivation for evaluating models, we added a few sentences to the introduction: "Though paleo-proxy records are an invaluable tool to reconstruct the climate system, they are sometimes scarce or completely absent over entire regions. Climate models can help fill these gaps, as they provide a full 3-dimensional and dynamically consistent representation of the climate system. However, as climate models are not perfect representations of Earth's climate, it is important to continually evaluate their performance."

(2) Closely linked to (1), the choice of models being evaluated is difficult to defend. Evaluating CMIP5-PMIP3 versus CMIP6-PMIP4 – and whether models have improved in their ability to simulation SO sea ice would seem valuable and more easy to understand as motivation. However the paper in its current form does not do this. Instead, it shows results from the older PMIP3 simulations, alongside some more recent LoveClim simulations. At least one of the LoveClim simulations also has a very strange seasonal
cycle of sea ice. The motivation for doing this approach, as opposed to PMIP3 versus PMIP4 is never explained. I would encourage the authors to consider focussing on PMIP4 versus PMIP3/2. More than 10 PMIP4 LGM simulations are available to the authors.

We understand and agree with the reviewer's point here. When we originally started the project (late Fall of 2018), the PMIP4 data was not yet available (it became available late 2020 / early 2021) which is why we decided to use PMIP3 data. However, in the new draft of this manuscript, we have included the available PMIP4 data and compared that against the PMIP3 outputs. Additionally, the strange seasonal

cycle in LOVECLIM1 (now referred to as weakNA) was a mistake. In the new draft, we are analyzing the LOVECLIM1 output with LOVECLIM1's summer months now correctly adjusted to February and March.

(3) The section on wind stress results is not particularly helpful. It needs more on how glacial-interglacial wind changes depend on model biases, and the sea ice itself. It can help to start with considered wind velocities above the surface, before then considering wind stress/curl changes. Otherwise results can tend to confuse.

We thank the reviewer for this comment. Our main goal in this section of our study is to understand the dynamic processes that lead to the simulated sea ice extent within each model. We want to understand how sea ice extent is related to wind stress within each of the models; i.e., in a numerical and theoretical world that is not necessarily realistic, but logical within itself. We are therefore not concerned about the ability of the models to accurately represent the winds, we want to know how the winds impact the sea-ice cover within the model framework. We agree that a thorough model-data comparison and evaluation of surface winds would be very interesting, but this is out of scope for this study. Please also refer to the minor comments section, where we provide a more detailed answer to this comment.

Minor comments

L90-_95. Be specific about what 'common' (e.g. 1 in 2 years?) and 'episodic' (e.g. 1 in 10 years, 1 in 50 years?) mean.

It is worth noting that geological data are generally extracted from 1 cm thick slices of sediment that average tens to hundreds of years of accumulation depending on the sedimentation rates. Therefore, though we understand the concern of the reviewer, the % of FCC or diatom transfer function data cannot provide quantitative recurrences.

We accordingly updated section 2.2 of the methods:
"Relative abundances of the indicator diatoms above 3% are thought to indicate the presence of sea ice over the core site (mean sea-ice extent north of the core site) while relative abundances between 1 and 3% suggest the episodic presence of sea ice over the core site (mean sea-ice edge south of the core site but maximum sea-ice edge north of the core site)".

Table 2 – It is most strange that LOVECLIM1 has a SIE minimum in Jan-Feb. Suggest excluding this model from the analysis. Hard to see how the results can be meaningful.

We apologize as there was a mistake in table 2. The SIE minimum in LOVECLIM1 is indeed Feb-Mar, similar to 8 of the other 9 models. This has been corrected in the revised manuscript.

L110 "The proxy"

This has been corrected.

Figure 1 – add latitude grid lines.

We have added latitude grid lines to Figure 1.

L160 onward. Given the freezing point of seawater is _-1.8C, I find the lack of <0C datapoints a little strange.

First, the paleo SST data is based on several approaches, including diatom transfer functions and radiolarian transfer functions (Gersonde et al., 2005). Micro-organisms develop during the sunlit period (spring to fall), when sea ice retreated and SSTs are back above the freezing point. Indeed, SST as low as -1.8° C would indicate yearly round sea-ice coverage during the whole LGM at that location. Such a situation would prevent any plankton production. This is why summer SST is the first explanatory variable for the distribution of many diatom species (Esper et al., 2014).

Second, modern diatom databases cover a SST range from -2°C to 20°C, winter sea-ice concentration from 100% to 0% and summer sea ice concentration from 40% to 0% (Armand et al., 2005; Crosta et al., 2005; Esper et al., 2010, 2014). It is probable that transfer functions would reconstruct < 0°C LGM SST, and high sea-ice concentrations, if adequate cores were available. For example, a diatom transfer function estimated LGM SST as low as -1°C at 60°S-5°W off the Weddell Sea (Gersonde et al., 2003). This core was not used in Gersonde et al. (2005), and therefore in the present compilation, because of its weak chronological constraint.

Would be useful to have some discussion of the relationship between the freezing point (_-1.8C) and the model and obs datapoints here.

We discuss this relationship more within section 3.3 of the results, shown below:

Results Section 3.3:

"Due to biological limitations, diatom transfer functions are mostly available in regions with low sea-ice cover. As such, our proxy summer SST compilation only contains two locations with SST temperatures below 0° C. With limited proxy SST data near the freezing point, we instead assess the model-data fit at the 1° isoline."

Figure 2. Consider adding a -1.8C or -2C line on panel (c).

We included a -2° C line in the revised Figure 2.

Figure 2. Obs data is too faint on panel (c). Make this more visible please.

This is noted and has been fixed.

Table 3 – order of columns seems strange. Maybe should be Winter: SI edge, SIE, SIA, Obs agreement, SST, SST agreement; then Summer: SI edge, SIE, SIA, Obs agreement, SST, SST agreement.

We agree with this suggestion. The table has been amended as suggested.

Figure 3 – Cannot see Southern Ocean temperature variation with current colorbar. Change scale to -2 to +8? Figure not useful with this colorscale.

We have fixed this by using a non-linear color bar.

Line 229-253 There are three main problems with this section.
Firstly it currently lacks adequate discussion of the fact that SHWs are too poorly simulated by most PMIP3 models to be able to straightforwardly interpret wind results.

We want to note that our main goal here is to analyse the internal dynamic processes within each model framework, to better understand summer sea ice extent, and what is driving this extent, within each simulation. Hence we were not particularly concerned about the ability of the models to accurately represent the winds. We have made this goal clearer but also highlighted the fact that there are significant uncertainties in both reconstructions and model simulations. The updated sections of our results and discussion are below.

Results section 3.4:
"While latitudinal position and magnitude of southern hemispheric westerlies at the LGM is poorly constrained (Kohfeld et al., 2013; Sime et al., 2016), here we want to assess the impact of the simulated windstress curl on ocean dynamics in each model. We thus use the simulated windstress outputs to estimate the location and strength of the SO upwelling, and its potential impact on sea-ice cover. "

Discussion:
"Given the uncertainties that surround the magnitude and the position of the Southern Hemisphere westerlies at the LGM (e.g., Kohfeld et al., 2013; Sime et al., 2016), this casts additional uncertainties on the location of the austral summer sea-ice edge."

Secondly, the discussion needs to take account of the control that sea ice exerts on SH winds.

We now also mention that the wind-sea ice relationship goes both ways, and the presence or absence of sea ice can also influence winds. We however want to reiterate that we are more interested in the impact of Southern Ocean upwelling (driven by SH winds) on sea ice. This is the reason we are looking at the windstress curl. The updated sentence is within section 3.3 of the results and included below in response to the next comment.

Thirdly, by focusing on stress and curl, rather than simpler above surface speed/velocity measures (which are less affected by surface roughness) it leads to a discussion that probably confuses the reader more than clarifying anything. Suggest re-reading relevant SO/sea ice wind papers by Bracegirdle et al (2013, 2018 and others), Sime et al (2013, 2016), and Kidston et al (2011, The influence of Southern Hemisphere sea-ice extent on the latitude of the mid-latitude jet stream, Geophys.
Res. Lett., 38, L15804, doi:10.1029/2011GL048056). Rewrite or remove this section?

We understand the point of the reviewer and agree that the cited studies highlight the impact of changes in Southern Ocean sea-ice extent on the position and strength of the SH westerlies. However, here we are interested in the impact of Southern Ocean upwelling, and associated divergence of surface waters, on sea ice. We have modified the caption of new figure 4 as well as the associated text as follows:

Results section 3.3:
"The strength and location of the southern hemispheric westerly and polar easterly winds impact Southern Ocean circulation, sea ice transport and therefore sea-ice distribution (Purich et al., 2016; Holland and Kwok, 2012). On the other hand, the presence or absence of sea ice also has a direct influence on surface winds (Kidston et al. 2011; Sime et al. 2016). The divergence created by the wind stress curl over the Southern Ocean leads to an upwelling of warmer deep waters and thus heat loss to the atmosphere. This upwelling can therefore also impact Southern Ocean sea-ice distribution."

---

## Author Comment (AC3) · 30 Nov 2021

We thank the reviewer for these useful comments which will certainly improve the manuscript. The reviewers' comments are reproduced in black, and our replies are provided in blue. When necessary, we have also included the modified section of the revised manuscript below our response.

**Review RC2 (10 February 2021)**

General Comments

This study compares PMIP3-simulated sea ice cover in the Southern Ocean, as well as that from two LOVECLIM experiments, against an updated catalogue of sea ice paleoproxy data. In that sense, it follows directly on Roche et al (2012) and Marzocchi and Jansen (2017). They focus on the summer season, as that season has had the fewest data constraints in previous studies. Additionally, they correlate the sea ice edge to SST and wind stress curl in the models.

Overall, I find the results of this study underwhelming. Although they provide a lot of results, I feel that not a lot of meaning is derived from them (i.e. Too much of a "figure tour" or "data tour").

We appreciate the comment and have now improved the manuscript to more clearly state our goals and results. Additionally, we have added PMIP4 simulations from six models to this intercomparison (plus an additional LOVECLIM simulation following the PMIP4 protocol), which will allow us to compare the LGM seasonal sea-ice cover differences between PMIP3 and PMIP4 models.

Assessing the paleo-proxy data is outside my area of expertise, but the model analysis does not feel like a substantial contribution beyond Marzocchi and Jansen (2017), who examined Southern Hemisphere sea ice controls in these same PMIP3 simulations.

We believe that our goals are different from the goal of Marzocchi and Jansen, as they analyse how sea ice impacts deep ocean circulation. Here, we want to i) assess the most likely seasonal sea-ice cover at the LGM by looking at both model simulations and proxies and ii) understand the processes that can impact the summer sea-ice distribution in the models.

In addition, our manuscript provides an updated compilation of sea ice and SST proxy data for the LGM, which is now more clearly stated. Furthermore, our study provides spatial comparisons between new compilations of proxy record estimates and simulated sea-ice extent and SST, which was not done in Marzocchi and Jansen (2017). Finally, as mentioned above, PMIP4 simulations are now also included in our study.

Indeed, I find it concerning that a serious flaw in one of those simulations reported in Marzocchi and Jansen (2017) (that of a bugrelated absence of wind-stress feedback on sea ice in

MRI-CGCM3) is not mentioned here, even though wind-stress curl is one of the foci of the study.

We thank the reviewer for pointing this out. We have now included a sentence about the bug in the MRI-CGCM3 model in the results section, seen below:

Results section 3.4:

"Our analysis suggests that the summer sea-ice edge in the MRI-CGCM3 is dynamically driven as its mean SO SST is close to the PMIP3 MMM, and its summer sea-ice edge is close to the maximum of the wind stress curl. However, this result should be taken with caution as in the MRI-CGCM3 simulation the coupling between sea ice and wind-stress at the ice/atmosphere interface was absent due to a model bug (Figure S4, Marzocchi & Jansen 2017)."

The two additional LOVECLIM simulations do not appear to be used in any significant way to get at causal mechanisms, even though these simulations differ substantially in both the design of the experiments and complexity of the model.

We acknowledge that there are experimental design differences between the LOVECLIM and PMIP3 model simulations presented here. We have thus removed the 2 LOVECLIM experiments from the PMIP3 group, and are presenting them separately. In line with other comments, we are now including PMIP4 LGM results, including one new simulation performed with LOVECLIM. We are thus now showing 3 different LOVECLIM simulations, in which the AMOC states differ. This allows us to assess inter and intra model differences, which is useful to understand the processes at play as well as to better constrain the LGM sea-ice extent. We also refer to this more in detail below in the Specific Comments section.

Finally, the authors do not provide much discussion of their results in the context of the previously-mentioned two papers, and given their claim that "the multi-model mean of austral summer and winter sea ice cover seem to provide good estimates of LGM conditions" appears to be at odds with the results presented in those papers for earlier generation (Roche et al, 2012) and the same model (Marzocchi and Jansen, 2017) data, I would have liked a bigger exploration of this difference.

Marzocchi and Jansen (2017) did not provide a multi-model mean of LGM sea-ice extent and did not show the spatial extent of the seasonal sea-ice in the models and proxy records. Roche et al. (2012) show the spatial extent of the seasonal sea-ice in the models and proxy records, but do not provide a multi-model mean and use results from PMIP2, whereas we are using the PMIP3 and PMIP4 LGM results. Finally, the paleo-data compilation that we are presenting includes all the recent Southern Ocean sea-ice records thus providing an updated view on proxy estimates of summer sea-ice extent compared to the two studies mentioned above. These differences will clearly be indicated in the Introduction of our revised manuscript.

Introduction:

"PMIP2 LGM simulations suggested that simulated LGM Antarctic sea-ice cover did not reflect the zonal variability nor the seasonality seen in proxy reconstructions (Roche et al, 2012). PMIP3 LGM simulations have also been analyzed, however not spatially, with results highlighting large inter-model differences in annual-mean, minimum and maximum Antarctic sea-ice area, and suggesting most PMIP3 models underestimate austral winter sea ice cover in comparison to proxy data (Sime et al., 2016, Marzocchi and Jansen 2017). Therefore, a spatial analysis of seasonal LGM sea ice simulated by PMIP3 models is lacking, as well as an analysis of LGM seasonal sea-extent simulated by a PMIP3 multi-model mean. Furthermore, no seasonal sea-ice analysis of PMIP4 simulations under LGM boundary conditions has been performed yet."

I found it hard to interpret these differences on my own due to the small size and indistinguishable lines in the figures and vague descriptions of methodologies.

We are unsure exactly what figures and which descriptions of the methods this is referring to, however, all figures have been significantly improved.

Specific Comments

intro raises connections w/ SAM – why not connect results to SAM?

The SAM mode of variability tells us about the strength and position of the westerlies in a non-periodic way over a wide range of timescales (generally from inter-annual to multi-annual). We don't believe it would make sense to connect this to our results since we are looking at the mean state of sea-ice concentration over centuries to millennia when other drivers played on the SO wind system.

Line 55: while Marzocchi and Jansen didn't focus on seasonality of PMIP3 simulations, they did plot the seasonal comparisons btw PI and LGM and discuss the characteristics of the seasonality in the models

To acknowledge the seasonal sea ice characteristics discussed in Marzocchi and Jansen we have added following sentence to our introduction, shown below:

Introduction:

" PMIP3 LGM simulations have also been analyzed, however not spatially, with results highlighting large inter-model differences in annual-mean, minimum and maximum Antarctic sea-ice area, in addition to most PMIP3 models underestimating austral winter sea ice cover in comparison to proxy data (Sime et al., 2016, Marzocchi and Jansen 2017)."

Within the introduction, we are now more explicit in placing our work into context with the prior related research. We provide these changes in the general comments section of this author response letter.

Given zonal asymmetries in sea ice edge, usefulness of hemispheric, zonal averages is unclear

We agree with the reviewer that there are some limitations when using zonal averages for analysis due to zonal asymmetries, but we cannot think of a better way to analyze the data.

Line 67-69: Multiple simulations from same model were averaged over – what if they involved different components? How reflective is the ensemble mean of the behaviour of any one simulation? (look into GISS-E2-R)

We thank the reviewer for pointing this out as we should have provided more information on these models in the manuscript. Three different models submitted two different LGM simulations (CCSM4, GISS-E2-R, MPI-ESM-P). The CCSM4 model runs differed in their initial states while the GISS-E2-R and MPI-ESM-P runs differed slightly in their physics. With both LGM runs yielding similar sea-ice extent for each of the three models and following Sime et al (2016) methodology, we decided to take an average of each model so that we would have one set of data per model.

We have now added the following sentences to the Methods section (lines 74-77):
"Three models submitted two different simulations to PMIP3 (CCSM4, GISS-E2-R, MPI-ESM-P). These simulations differed because of a difference in the initial state (CCSM4), or small changes in the physics of the model (GISS-E2-R and MPI-ESM-P, described in Table 1 of Kageyama et al. 2021). Following Sime et al. (2016), we chose to average the simulations for the models who submitted two LGM runs, yielding one output per model."

The LOVECLIM simulations that were chosen are not representative of the PMIP3 models for a number of reasons, and the differences in their attributes, the reasons for their selection and the implications for the results are not discussed much here. As a result, the comparison feels artificial and not very instructive. Looking further into the simulations in Menviel et al (2017), I can make a guess as to why these ones were chosen, on the basis of the performance of their carbon cycle models. However, the fact that these simulations were performed with prognostic $CO_2$ concentrations (via a carbon cycle model ) rather than prescribed emissions/concentrations, and were spun up in a transient fashion from 35ka BP rather than just equilibrated to fixed LGM conditions as most of the PMIP3 simulations would have been, and had anomalous hosing applied in either or both of the Northern and Southern Hemispheres has bearing on the interpretation of the results. However, only the hosing is mentioned, and the implications of these design choices on the sea ice distributions are not discussed.

We understand the concerns of the reviewer and now provide a better motivation to include the LOVECLIM simulations. Given their skills in representing the LGM oceanic tracer distributions, it is interesting to include LOVECLIM1 (now referred to as weakNA) and LOVECLIM2 (now referred to as weakNA_AB) in the inter-model comparison. These simulations are however not included in the PMIP3 pool of simulations. In addition, we are now presenting a LOVECLIM simulation, which follows the PMIP4 protocol. These three LGM simulations performed with LOVECLIM allow us to compare the inter-model to intra-model range in sea-ice extent. These three simulations, which feature very different oceanic circulation states, highlight that the inter-model spread is much smaller than the intra-model spread, suggesting that an improved model-data fit within one model framework cannot provide (much) information on the oceanic circulation state.

I am concerned about the fact that at least one lower-resolution model (e.g. LOVECLIM, whose ocean is nominally 3degx3deg) was interpolated onto a higher-resolution grid (1degx1deg) for plotting the results. This artificially inflates the apparent resolution of the results and thus encourages attempts to interpret changes at smaller spatial scales than the model provides as real. Whether the resolution of any PMIP3 models was artificially inflated in this way is not clear.

For some aspects of our analysis, we need all the models to be on the same grid. For the figures where this is not necessary (Figure 3 and Figure 4) we will plot the results in their original grids in the revised manuscript.

Lines 82-97 I am not a specialist in the interpretation of sea ice proxy records, so I found this section confusing. My main source of confusion lay in interpreting the uncertainties related to the apparently weak signals (differences between 1-3% in diatom assemblages), given the RMSEP values were 10%. I'm assuming the two percentages were not referring to the same quantities. I'm not expecting this paper to provide an overview of this method, assuming the references already provide that, but a sentence or two to make these results interpretable to non-specialists would be appreciated.

Indeed, the reviewer rightly understood that these are different quantities referring to different metrics. First, Gersonde and Zielisnski (2000) defined two qualitative metrics to infer the past position of winter and summer sea-ice edges. Using sediment trap series and core-tops in the Atlantic sector of the Southern Ocean they showed that >3% of *Fragilariopsis curta+cyclindrus* and >3% of *F. obliquecostata* best track the mean position of winter and summer edges, respectively. Although these quantities appear low, it is worth noting that these species thrive only at very low SSTs (-1 to 1°C) and high sea-ice concentrations (> 70% of WSIC; Armand et al., 2005) and are generally not present (0%) in the open ocean. Second, Crosta et al. (1998) developed a diatom-based transfer function that uses a greater number of sea-ice-related species. It allows us to quantify sea-ice duration or sea-ice concentration with an error on the calibration step, which represents the capacity of the transfer function to reconstruct the distribution of the modern fields of the parameters (here winter and summer sea-ice concentrations). This error is around 1 month per year for sea-ice duration and ~10% for sea-ice concentration.

We tried to rephrase and simplify the whole paragraph to make it better understandable to non-specialists:

Methods Section 2.2:

"The numerical simulations are compared to a compilation of 149 proxy records covering the LGM (See Table S1 in the Supplement, Allen et al. 2011; Benz et al. 2016; Ferry et al. 2015; Xiao et al. 2016; Gersonde et al. 2005; Ghadi et al. 2020; Nair et al. 2019). Quantitative SST was reconstructed at 138 locations, proxies for winter sea-ice presence or concentration were available at 149 locations and proxies for summer sea-ice presence were available at 132 locations. SSTs were derived from diatom-based transfer functions (Crosta et al., 1998; Esper and Gersonde, 2014a) while winter and summer sea-ice extent were derived either from the relative abundance of sea-ice indicator diatoms, respectively the *Fragilariopsis curta* group and *F.obliquecostata* (Gersonde et al., 2005), or diatom-based transfer functions whenever possible (Crosta et al., 1998; Esper et al., 2014b). Relative abundances of the indicator diatoms greater than 3% are thought to indicate the common presence of sea ice (average sea-ice extent) while relative abundances between 1 and 3% suggest the episodic presence of sea ice (maximum sea-ice extent). In this study, we characterize the relative abundance of >3% as evidence of sea ice and the relative abundance between 1 and 3% as evidence for possible sea ice presence over the core sites during the LGM. Quantitative values were considered to indicate the presence of winter sea ice when they were above the root mean square error of prediction (RMSEP) on the validation models, generally around 10% for winter sea ice (Crosta et al., 1998; Esper et al., 2014b). It is however worth noting that quantitative values were always below the RMSEP of ~10% for summer sea ice in the validation model, therefore suggesting only episodic presence of summer sea ice over a restricted number of sites."

Why were two months selected to define sea ice maxima and minima ? Was this based on prior knowledge, or analyses of the seasonal cycle in the models?

We chose two months to define sea-ice maxima and minima because when looking at the sea ice output, some of the models had minima and maxima that persisted longer than one month. We have gone back and checked the sea-ice extent for the PMIP3 models. The difference between the one-month minima and the two-month minima is relatively small and does not affect our conclusions (see table below):

| PMIP3 models | 2 month average summer SIE $(10^6 \, km^2)$ | 1 month average summer SIE $(10^6 \, km^2)$ |
| --- | --- | --- |
| CNRM | 0.06 | 0.0049 |
| GISS-E2-R | 2.39 | 1.92 |
| IPSL-CM5A-LR | 2.41 | 2.14 |
| MIROC-ESM-P | 3.53 | 2.86 |
| MPI-ESM-P | 4.48 | 4.17 |
| MRI-CGCM3 | 12.54 | 11.98 |
| FGOALS-G2 | 13.62 | 10.66 |

| | | |
|---|---|---|
| CCSM4 | 27.46 | 25.40 |

| PMIP4 Models | 2 month average summer SIE ($10^6$ km$_2$) | 1 month average summer SIE ($10^6$ km$_2$) |
|---|---|---|
| MIROC-ES2L | 0.4 | 0.14 |
| IPSL-CM5A2 | 2.48 | 2.29 |
| MPI-ESM1-2 | 3.64 | 3.16 |
| AWI-ESM-1 | 14.66 | 14.32 |
| LOVECLIM | 15.54 | 14.31 |
| CESM1.2 | 23.63 | 21.42 |
| UoT-CCSM4 | 33.45 | 32.33 |

To reflect this, we have added in the method that using 2 months-mean leads to a larger summer sea-ice extent that the one that would be obtained from a 1 month mean. However, we believe that the 2 months-mean is more appropriate for a comparison with proxy records.

Lines 104-106 I'm not entirely clear on the methodology here. Firstly, were zonal averages performed in each ocean basin region over the latitudes of 15% sea ice concentrations for each longitude division (= 1 grid cell) or over sea ice concentrations in each latitude band (from which the 15% was then calculated)?

We zonally averaged over each latitude band, then determined at which latitude the models reached a 15% sea-ice concentration. For the models that did not reach 15% sea-ice concentration at any latitude after the full zonal average, we then zonally averaged over each ocean basin and determined the latitude (within that ocean basin) in which the model reached 15% sea-ice concentration. We amended the manuscript as follows:

Methods section 2.3:

"The sea-ice edge is defined as the 15% sea-ice concentration isoline. We calculate this by zonally averaging across all longitudes for each latitude band, then determining at which latitude the model simulates a minimum of 15% sea-ice concentration. For model simulations that do not reach 15% of sea-ice concentration in some regions of the Southern Ocean, we average over the regions with sufficient sea-ice cover only, and classify the resulting latitude as the sea-ice edge for the model."

And then, when defining an hemispheric sea ice latitude, am I correct in understanding that a zonal average over all longitudes was not calculated and instead, an average (weighted or unweighted? ) was performed over the individual ocean basin regions? If this is correct, why was this method chosen?

We calculated a zonal average over all longitudes. This was made more clear in the modified methods section highlighted above.

Lines 118-119 Why does the multi-model mean lead to an asymmetric region of variance north and south of the mean lines? Are you suggesting that the simulations are distributed in a non-Gaussian way? Based on visual inspection, I wonder how the mean was calculated. I'm assuming the multi-model mean was calculated by averaging over the latitude of sea ice margin for each cell's longitude range from each run, but it doesn't seem to match what I see in the figure. For example, between 150 and 180degE, two, maybe 3, runs extend past the latitude of the outermost blue points. That leaves 6ish runs south of those points, but the multi-model mean lies north of the points. If the two runs were far north of the points, I would understand this, but that is not the case. Rereading the text, I wonder if the authors are suggesting they performed a multi-model average of sea ice concentration in every grid cell and then calculated the 15% concentration margin from the ensemble-mean distribution.

This is correct. We calculated the mean sea-ice concentration among all the models in every grid cell, then calculated the 15% concentration isoline of that multi-model mean. As mentioned in your next comment, models with high sea-ice concentration up until their margin can pull this mean in one way or the other, and therefore the multi-model mean does not fall perfectly in the middle between the models.

This is now clearly indicated in section 2.3 of our methods:

"To calculate the multi-model mean (MMM), we average sea-ice concentration over each grid cell for all models (PMIP3, PMIP4, and LOVECLIM sensitivity runs separately). We then calculate the 15% sea-ice concentration isoline of each multi-model mean."

If those more extensive runs had very high sea ice concentration up until their margins, it might explain the position of the mean, but it's not clear to me how the standard deviations of the 15% line could be derived from this calculation. Since assigning a cutoff to sea ice at the 15% concentration is a non-linear operation, I would expect to get a more Gaussian multi-model distribution for the latitudes of the 15% lines than performing the calculation on the ensemble means of the sea ice concentrations directly.

Similarly, we calculate the standard deviation in each individual grid cell. Each model and MMM has a sea ice concentration value for each individual grid cell, and therefore we can simply calculate the standard deviation for each individual grid cell.

This is also clarified in section 2.3 of our methods:

"To calculate the standard deviation, we similarly compute a standard deviation value for each individual grid cell, before adding and subtracting that standard deviation (σ) of sea-ice concentration from the MMM for each grid cell. The +/- 1σ then represents the 15% sea-ice concentration isoline calculated from the MMM +/- 1σ. Notably, this creates a non-symmetric standard deviation isoline as each grid cell has its own MMM (and σ) value, calculated independently from any surrounding grid cells."

Given the zonal asymmetries in the summer data and their relative absence in the models (based on inspection of the multi-model mean in Figure 1 and the performance of PMIP2 models in Roche et al, 2012), I'd like to see the analysis performed for Figure 2 calculated on a regional basis, rather than based on hemispheric averages. Such an analysis would be more likely to bring out discrepancies between the models and the data than the hemispheric average.

We understand the comment of the Reviewer, however given the limited proxy data and the extent of our analysis, which includes both zonal averages and spatial correlations (e.g. Figures 1 and 3, Table 3), we think that a regional assessment of SST vs sea-ice extent would not add much value.

Technical Comments

Two different definitions for LGM time period provided in the abstract and text

This has been corrected.

I find it very difficult to distinguish between the colours of the different lines in Figure 1, because they are so thin. If thickening was not chosen because of confusion in the plot, I think the line labels can easily be dropped to simplify the figure.

We have thickened the contour lines and have gotten rid of the line labels to make the contour lines more clear and distinguishable.

Also, without any latitudes and longitudes marked on the plot and only the southern tip of S. America included as a georeference, it takes a lot of work for someone who is not intimately familiar with Antarctic geography to translate the descriptions in the text (most of which seemed to be in longitudes) to their locations on the figure and compare the latitude values stated in the text with those in the figure.

We thank the reviewer for pointing this out. We have included the latitude and longitude gridlines and labels in Figure 1.

It would be helpful for the boundaries of the ocean basin regions to be marked in one figure. Their names are clear, but precisely where the boundaries between the basins are drawn and their northern and southern extents would be helpful in interpreting the results.

We thank the reviewer for pointing this out. We have included labels that clearly indicate each ocean basin in Figure 1.

Line 110: Sentence fragment "The proxy"

This has been corrected.

Line 214: Typo MRI-ESM-P should be MRI-CGCM3

This has been corrected.

---

## Author Comment (AC4) · 30 Nov 2021

We thank the reviewer for these useful comments which will certainly improve the manuscript. The reviewers' comments are reproduced in black, and our replies are provided in blue. When necessary, we have also included the modified section of the revised manuscript below our response.

**Review RC3 (12 April 2021)**

Green et al., propose some interesting insights related to seasonal changes in sea-ice cover during the LGM using PMIP3 and LOVECLIM simulations, in particular in relation to wind stress and surface ocean temperatures. While simulations are generally coherent with paleo reconstructions for LGM winter sea-ice cover, simulated glacial summer sea-ice cover differs widely between models and between models and (arguably limited) proxy-based reconstructions. While study is certainly relevant, it lacks some context, I feel. To me, the main outcome of the present study relates to the conclusion that the seasonal contrast in sea-ice cover is reduced today compared to the LGM. While clearly interesting, this finding should be placed into a broader context and outline how increased sea-ice dynamics would affect ocean circulation and more generally carbon and nutrient biogeochemistry during the LGM. Furthermore, it remains unclear why reconstructing summer sea-ice extent is relevant, in particular for air-sea gas exchange and deep water production, which to a large degree occur in winter. I certainly support publication of the present study in Climate of the Past, provided some additional contextualization can be provided.

We thank the reviewer for these comments and will include more information regarding the motivation of our work in addition to highlighting the broader contexts of our results. To clarify, through providing a spatial and seasonal assessment of LGM Southern Ocean sea-ice extent in PMIP3 models (and now also PMIP4 models), and then comparing that output to available sea-ice and SST records, we believe the main outcome of this manuscript is providing a better constraint on summer sea-ice extent, which is a slightly larger estimate than what has previously been published (Lhardy et al. 2020, Roche et al. 2012).

Our goals are now clarified in the Introduction and shown in the first response in the General Comment section below.

Additionally, our main outcome is clarified in the first paragraph of the discussion:

"We suggest that during the LGM, the likely austral winter sea-ice edge was between 50.5˚S and 51.5˚ S and during austral summer the sea-edge was likely between 61˚ and 62˚ S, with a mean sea-ice extent of 14-15 x $10^6$ km$^2$, similar to the sea ice characteristics simulated by AWI-ESM-1 and FGOALS-G2. This is an improved constraint on LGM summer sea-ice extent as we use modeled sea ice in conjunction with paleo proxy SST and sea ice data to calculate our estimate. Our LGM summer sea-ice extent estimate is slightly larger than previous estimates of ~10.2 x $10^6$ km$^2$ (Lhardy et al., 2021) and ~11.1 x $10^6$ km$^2$ (Roche et al., 2012), which only use paleo proxy data to calculate their estimates."

It is important we reconstruct summer sea ice extent due to the impact of seasonal sea ice changes on Southern Ocean dynamics. We have included a sentence within our introduction highlighting this:

Introduction:
"Specifically, seasonal changes in sea ice have large implications on Southern Ocean dynamics through buoyancy (Marzocchi and Jansen, 2017) and carbon cycle changes (Haumann et al., 2016). Understanding these past changes in sea ice and their natural drivers, at different timescales and under different boundary conditions, will therefore allow us to better project future sea ice changes."

General comment

I'm certainly not a modeling expert, but the reason underlying the poor representation of the glacial summer sea-ice extent should be clarified. In particular, I don't quite understand why the distribution of mean summer SST during the LGM is generally coherent between models while summer sea-ice extent is not. Maybe this aspect could be clarified further at least for non-specialized readers.

Our study tries to provide answers to the questions raised by the reviewer, namely: is the LGM seasonal sea-ice extent coherent across models? Why are there such large differences? Can we provide constraints on the actual LGM seasonal sea-ice extent?
However, contrary to the reviewer's comment and as shown in Figure 2, the distribution of mean summer SST is not coherent across the models, with a ~3˚ C spread at 75˚ S and almost 6˚ C spread at 65˚ S. The fact that models display different ocean temperatures in the Southern Ocean will be clarified in the revised manuscript. Similarly, the suggested relationship between summer sea-ice extent and seawater temperature will also be clarified.

We are now more clearly defining the goals of the study in the Introduction:
"Here, we assess the minimum and maximum Southern Ocean sea-ice covers as simulated in LGM PMIP3 and PMIP4 experiments. To better assess intra- versus inter-model variability, LGM sensitivity experiments performed with LOVECLIM are also included. The PMIP3, PMIP4 and LOVECLIM experiments are compared to available LGM sea-ice and SST paleo-proxy data, allowing us to determine the best performing models. Combining models and proxy data, we can provide an updated estimate of seasonal Southern Ocean sea-ice cover during the LGM. Furthermore, we analyze the processes that lead to the inter-model differences in summer sea-ice extent at the LGM."

We are also more clearly mentioning the spread of simulated summer SSTs between the models when discussing Figure 2:

"Among the models, the distribution of zonally averaged SSTs are not consistent over the Southern Ocean. For example, at 75˚ S both PMIP3 and PMIP4 models simulate a SST spread of ~3˚ C while at 65˚ S both model groups simulate a SST spread of ~6˚ C (Figure 2c,d)."

Detailed comments

l. 15-27. Since the MS is focusing on seasonal contrasts in sea-ice extent, it may be relevant to briefly explain how seasonality affects the main processes outlined in this section today. For example, does a positive SAM phase affect both the winter and summer sea-ice extent linearly?

The reviewer makes a good point here. We are now describing the impact of SAM on seasonal sea-ice extent in the introduction. Since 1970 there has been a positive trend in the SAM, particularly in austral summer. Doddridge and Marshall (2017) found that a positive SAM led to a high latitude SST decrease and sea-ice increase with a damping timescale of ~3 months, except around the Antarctic Peninsula where the response of sea ice to positive SAM is toward a decrease. However, Ferreira et al. (2015) suggested that the short-term and long-term response to changes in the Southern Hemispheric westerlies are different, with positive SAM trends leading to a short-term cooling but a long-term warming due to the upwelling of relatively warm circumpolar deep waters. Geological data embed several tens of years of sedimentation. We are here looking at the LGM mean state and therefore also the mean response to the average position of the Southern Hemispheric westerlies over centuries to millennia.

Introduction:
"Doddridge and Marshall (2017) have shown that, on a seasonal timescale, Southern Ocean sea-ice was responding to changes in the southern annual mode (SAM), with a positive phase of the SAM leading to lower Southern Ocean sea surface temperatures (SST) and a larger sea-ice extent. However, on longer timescales, a positive phase of the SAM can lead to a Southern Ocean warming due to the enhanced upwelling of relatively warm circumpolar deep waters (Ferreira et al. 2015). "

l. 27-30. The statement is misleading. I believe the Southern Ocean has accounted for about 40% of the global OCEANIC uptake of anthropogenic CO2.

The reviewer is correct here and this will be fixed. The updated sentence is below:

Introduction:
"Given that the Southern Ocean has accounted for ~40% of the oceanic anthropogenic CO2 uptake between 1870 and 1995 (Sabine et al., 2004; Frölicher et al., 2015; Mikaloff-Fletcher et al., 2006; Landschützer et al., 2015, Watson et al., 2020), it is crucial to better understand the processes that impact Antarctic sea-ice cover."

l. 44. Could you briefly explain why records of summer sea-ice extent are generally more poorly constrained?

We decided to cut this sentence from our introduction, and the updated paragraph is below. However, for your information, most of the sediment cores from the Southern Ocean are from "topographic highs" such as the mid-oceanic ridges and Antarctic coast and slope. In between, the abyssal plains reach down to 5000m deep, where sedimentation rates are extremely low, microfossils are rare, dissolution is important

and chronological issues are predominant. Though WSI reached the mid-oceanic ridges (several cores show it), SSI probably laid over the abyssal plains (so very difficult to get records of).

Introduction:
"Antarctic sea ice at the LGM was reconstructed using proxy data for the first time in 1981 (CLIMAP-Project-Members,1981). The LGM sea-ice compilation covering the Southern Ocean of Gersonde et al. (2005) is routinely used to provide estimates of LGM sea-ice cover. New Southern Ocean sea-ice data, reconstructed using statistical methods and diatom assemblages, has however been published since then (Allen et al., 2011; Benz et al., 2016; Ferry et al., 2015; Xiao et al., 2016; Ghadi et al., 2020; Nair et al., 2019) thus highlighting the need for an updated compilation. Here, we use this improved compilation, containing sea-ice records as well as summer SST estimates when present, to better constrain the minimum and maximum LGM sea-ice cover."

l. 110 – incomplete sentence

This has been fixed.

l. 161-163 & 168-169 – how does the relationship between SST and sea-ice cover equate for the winter months as a comparison?

There are similar relationships between ocean temperatures (SST and average temperatures across the entire water column) and sea-ice cover for both austral winter and summer. We have now included two new figures to our supplement showing both relationships and adjusted the Results section in the manuscript.

Results section 3.3:
"We first assess the relationship between zonally averaged austral summer SSTs in the Southern Ocean (50˚ S and 75˚ S) and sea-ice edge and extent (Figure 2a,b). The relationship between simulated summer SO SST and sea-ice edge or extent can be approximated by a linear fit, with R2 values of 0.90 and 0.81, respectively (Figure 2a, b, S2). Similarly, this relationship is also seen during austral winter. Using a linear fit to approximate this SST vs sea-ice edge and extent relationship we find R2 values of 0.80 and 0.88 (Figure S3)."

[Figure]

Figure R1 (referred to in the text as S2). The left column shows the relationship between austral summer SO (75˚S - 50˚S) SST and austral summer sea-ice edge (a) and sea-ice extent (c). The right column shows the relationship between annual mean SO temperatures averaged over the entire water column and austral summer sea-ice edge (b) and sea-ice extent (d). Using a linear fit approximation, the correlation for each relationship is found above each individual panel.

[Figure]

Figure R2 (referred to in the text as S3). The left column shows the relationship between austral winter SST (75°S - 50°S) and austral summer sea-ice edge (a) and sea-ice extent (c). The right column shows the relationship between annual mean SO temperatures averaged over the entire water column and austral summer sea-ice edge (b) and sea-ice extent (d). Using a linear fit approximation, the correlation for each relationship is found above each individual panel.

l. 203 – how does this value compare with modern SSI extent?

The modern average SSI extent for 1981-2010 is $3.1 \times 10^6$ km$^2$ (Eayrs et al., 2019). We now include a comparison with modern values in the first paragraph of our discussion:

Discussion:
" This can be compared to the average modern austral winter sea-ice extent of $18.5 \times 10^6$ km$^2$ and the average modern austral summer sea-ice extent of $3.1 \times 10^6$ km$^2$ (Eayrs et al., 2019)."

l. 321-323 – assuming that most of the mixing occurs during the winter, I'm not too sure to understand how increased sea-ice melt during the spring-summer could enhance nutrient utilization (and by inference carbon drawdown)?

Increased sea-ice melt leads to increased surface stratification (Galbraith and de Lavergne, 2019). Increased surface stratification reduces nutrient supply to the surface waters in which the phytoplankton thrives. It also allows for more time for nutrient uptake. Both processes lead to increased nutrient utilization at high latitudes (François et al., 1997; Sigman and Boyle 2000, Abelmann et al 2015), where much of the current nutrient supply goes unused (Sigman et al., 1999). This enhanced nutrient utilization leads to higher rates of carbon drawdown (Sigman et al., 2021).

---

## Author Comment (AC8) · 24 Jan 2022

We thank the reviewer for these useful comments which will certainly improve the manuscript. The reviewers' comments are reproduced in black, and our replies are provided in blue. When necessary, we have also included the modified section of the revised manuscript below our response.

**Review RC3 (12 April 2021)**

Green et al., propose some interesting insights related to seasonal changes in sea-ice cover during the LGM using PMIP3 and LOVECLIM simulations, in particular in relation to wind stress and surface ocean temperatures. While simulations are generally coherent with paleo reconstructions for LGM winter sea-ice cover, simulated glacial summer sea-ice cover differs widely between models and between models and (arguably limited) proxy-based reconstructions. While study is certainly relevant, it lacks some context, I feel. To me, the main outcome of the present study relates to the conclusion that the seasonal contrast in sea-ice cover is reduced today compared to the LGM. While clearly interesting, this finding should be placed into a broader context and outline how increased sea-ice dynamics would affect ocean circulation and more generally carbon and nutrient biogeochemistry during the LGM. Furthermore, it remains unclear why reconstructing summer sea-ice extent is relevant, in particular for air-sea gas exchange and deep water production, which to a large degree occur in winter. I certainly support publication of the present study in Climate of the Past, provided some additional contextualization can be provided.

We thank the reviewer for these comments and will include more information regarding the motivation of our work in addition to highlighting the broader contexts of our results. To clarify, through providing a spatial and seasonal assessment of LGM Southern Ocean sea-ice extent in PMIP3 models (and now also PMIP4 models), and then comparing that output to available sea-ice and SST records, we believe the main outcome of this manuscript is providing a better constraint on summer sea-ice extent, which is a slightly larger estimate than what has previously been published (Lhardy et al. 2020, Roche et al. 2012).

Our goals are now clarified in the Introduction and shown in the first response in the General Comment section below.

Additionally, our main outcome is clarified in the first paragraph of the discussion, lines 309-314:

"We suggest that during the LGM, the likely austral WSI edge was between 50.5˚S and 51.5˚S, with a mean sea-ice extent of 35-36 x $10^6$ km$^2$. During austral summer we suggest the sea-edge was likely between 61˚ and 62˚S, with a mean sea-ice extent of 14-15 x $10^6$ km$^2$, similar to the sea ice characteristics simulated by AWI-ESM-1 and FGOALS-G2. This is an improved constraint on LGM SSI extent as we use modeled sea ice in conjunction with paleo-proxy SST and sea ice data to calculate our estimate. Our LGM SSI extent estimate is slightly larger than previous estimates of ~ 10.2 x $10^6$ km$^2$ (Lhardy et al., 2021) and ~ 11.1 x $10^6$ km$^2$ (Roche et al., 2012), which only use sea-ice proxy data to calculate their estimates."

It is important we reconstruct summer sea ice extent due to the impact of seasonal sea ice changes on Southern Ocean dynamics. We have included a sentence within our introduction highlighting this:

Introduction lines 34-37:

"Specifically, seasonal changes in sea ice can significantly affect SO dynamics through buoyancy (Marzocchi and Jansen, 2017) and lead to changes in the atmosphere-ocean carbon exchange (Haumann et al., 2016). Understanding changes in sea ice and their natural drivers at different timescales and under different boundary conditions will allow us to better project future sea ice changes."

General comment

I'm certainly not a modeling expert, but the reason underlying the poor representation of the glacial summer sea-ice extent should be clarified. In particular, I don't quite understand why the distribution of mean summer SST during the LGM is generally coherent between models while summer sea-ice extent is not. Maybe this aspect could be clarified further at least for non-specialized readers.

Our study tries to provide answers to the questions raised by the reviewer, namely: is the LGM seasonal sea-ice extent coherent across models? Why are there such large differences? Can we provide constraints on the actual LGM seasonal sea-ice extent?
However, contrary to the reviewer's comment and as shown in Figure 2, the distribution of mean summer SST is not coherent across the models, with a ~3° C spread at 75° S and almost 6° C spread at 65° S. The fact that models display different ocean temperatures in the Southern Ocean will be clarified in the revised manuscript. Similarly, the suggested relationship between summer sea-ice extent and seawater temperature will also be clarified.

We are now more clearly defining the goals of the study in the Introduction, lines 64-69:

" Here, we assess the minimum and maximum SO sea-ice extent as simulated in LGM PMIP3 and PMIP4 experiments. To better assess intra- versus inter-model variability, a suite of LGM sensitivity experiments performed with the LOVECLIM model of intermediate complexity are also included. The PMIP3, PMIP4 and LOVECLIM experiments are compared to available sea-ice and SST paleo-proxy data, allowing us to determine the best model-data fit. Combining models and proxy data, we can provide an updated estimate of seasonal SO sea-ice cover during the LGM. Furthermore, we analyze the processes that lead to the inter-model differences in summer sea-ice extent at the LGM."

We are also more clearly mentioning the spread of simulated summer SSTs between the models when discussing Figure 2 in section 3.3 of the Results, lines 237-239:

"Among the models, the distribution of zonally averaged SSTs is not consistent over the SO. For example, at 75˚S both PMIP3 and PMIP4 models simulate a SST spread of SST around ~ 3˚C while at 65˚S both model groups simulate a SST spread of ~ 6˚C (Figure 2c, d)."

Detailed comments

l. 15-27. Since the MS is focusing on seasonal contrasts in sea-ice extent, it may be relevant to briefly explain how seasonality affects the main processes outlined in this section today. For example, does a positive SAM phase affect both the winter and summer sea-ice extent linearly?

The reviewer makes a good point here. We are now describing the impact of SAM on seasonal sea-ice extent in the introduction. Since 1970 there has been a positive trend in the SAM, particularly in austral summer. Doddridge and Marshall (2017) found that a positive SAM led to a high latitude SST decrease and sea-ice increase with a damping timescale of ~3 months, except around the Antarctic Peninsula where the response of sea ice to positive SAM is toward a decrease. However, Ferreira et al. (2015) suggested that the short-term and long-term response to changes in the Southern Hemispheric westerlies are different, with positive SAM trends leading to a short-term cooling but a long-term warming due to the upwelling of relatively warm circumpolar deep waters. Geological data embed several tens of years of sedimentation. We are here looking at the LGM mean state and therefore also the mean response to the average position of the Southern Hemispheric westerlies over centuries to millennia.

Introduction lines 25-29:
"Doddridge and Marshall (2017) have shown that, on a seasonal timescale, SO sea-ice was responding to changes in the southern annual mode (SAM), with a positive phase of the SAM leading to lower SO sea surface temperatures (SST) and a larger sea-ice extent. However, on longer timescales, a positive phase of the SAM can lead to a SO warming due to the enhanced upwelling of relatively warm circumpolar deep waters (Ferreira et al. 2015)."

l. 27-30. The statement is misleading. I believe the Southern Ocean has accounted for about 40% of the global OCEANIC uptake of anthropogenic CO2.

The reviewer is correct here and this will be fixed. The updated sentence is below:

Introduction lines 31-34:
"Given that the SO has accounted for ~ 40% of the oceanic anthropogenic CO2 uptake between 1870 and 1995 (Landschützer et al., 2015; Sabine et al., 2004; Frölicher et al., 2015; Mikaloff-Fletcher et al., 2006; Sabine et al., 2004; Watson et al., 2020), it is crucial to better understand the processes that impact Antarctic sea-ice cover."

l. 44. Could you briefly explain why records of summer sea-ice extent are generally more poorly constrained?

We decided to cut this sentence from our introduction, and the updated paragraph is below. However, for your information, most of the sediment cores from the Southern Ocean are from "topographic highs" such as the mid-oceanic ridges and Antarctic coast and slope. In between, the abyssal plains reach down to 5000m deep, where sedimentation rates are extremely low, microfossils are rare, dissolution is important and chronological issues are predominant. Though WSI reached the mid-oceanic ridges (several cores show it), SSI probably laid over the abyssal plains (so very difficult to get records of).

Introduction lines 44-49:
"While LGM Antarctic sea ice was first reconstructed in 1981 (CLIMAP-Project-Members, 1981), the proxy compilation of Gersonde et al. (2005) is nowadays routinely used to provide estimates of LGM sea-ice cover. Since 2005, additional SO sea-ice data has been published (Allen et al., 2011; Benz et al., 2016; Ferry et al., 2015; Ghadi et al., 2020; Nair et al., 2019; Xiao et al., 2016), and recently merged into an updated compilation (Lhardy et al., 2021). Within this updated sea-ice compilation, certain cores also contain summer SST estimates. We use this sea-ice proxy data along with the summer SST proxy data to better constrain the minimum and maximum LGM sea-ice cover."

l. 110 – incomplete sentence

This has been fixed.

l. 161-163 & 168-169 – how does the relationship between SST and sea-ice cover equate for the winter months as a comparison?

There are similar relationships between ocean temperatures (SST and average temperatures across the entire water column) and sea-ice cover for both austral winter and summer (Figure R1, R2). We have now included two new figures to our supplement showing both relationships and adjusted the Results section in the manuscript.

Sections 3.3 of Results, lines 221-226:

"We first assess the relationship between zonally averaged simulated austral summer SSTs in the SO (50˚S and 75˚S) and the simulated sea-ice edge and extent (Figure 2a, b). The relationship between simulated summer SO SST and SSI edge or extent can be approximated by a linear fit, with $R^2$ values of 0.90 and 0.81, respectively (Figure 2a, b). Similarly, this relationship is also seen during austral winter with $R^2$ values of 0.80 and 0.88 for the relationship between SST and sea-ice edge and extent, respectively (Figure 5)."

[Figure]

Figure R1. The left column shows the relationship between austral summer SO (75˚S - 50˚S) SST and austral summer sea-ice edge (a) and sea-ice extent (c). The right column shows the relationship between annual mean SO temperatures averaged over the entire water column and austral summer sea-ice edge (b) and sea-ice extent (d). Using a linear fit approximation, the correlation for each relationship is found above each individual panel.

[Figure]

Figure R2. The left column shows the relationship between austral winter SST (75˚S - 50˚S) and austral summer sea-ice edge (a) and sea-ice extent (c). The right column shows the relationship between annual mean SO temperatures averaged over the entire water column and austral summer sea-ice edge (b) and sea-ice extent (d). Using a linear fit approximation, the correlation for each relationship is found above each individual panel.

l. 203 – how does this value compare with modern SSI extent?

The modern average SSI extent for 1981-2010 is $3.1 \times 10^6$ km$^2$ (Eayrs et al., 2019). We now include a comparison with modern values in the first paragraph of our discussion:

Discussion lines 314-316:
"Our estimates can be compared to the average modern austral WSI extent of $18.5 \times 10^6$ km$^2$ and the average modern austral SSI extent of $3.1 \times 10^6$ km$^2$ (Eayrs et al., 2019)."

l. 321-323 – assuming that most of the mixing occurs during the winter, I'm not too sure to understand how increased sea-ice melt during the spring-summer could enhance nutrient utilization (and by inference carbon drawdown)?

Increased sea-ice melt leads to increased surface stratification (Galbraith and de Lavergne, 2019). Increased surface stratification reduces nutrient supply to the surface waters in which the phytoplankton thrives. It also allows for more time for nutrient uptake. Both processes lead to increased nutrient utilization at high latitudes (François et al., 1997; Sigman and Boyle 2000, Abelmann et al 2015), where much of the current nutrient supply goes unused (Sigman et al., 1999). This enhanced nutrient utilization leads to higher rates of carbon drawdown (Sigman et al., 2021).

---

## Author Response (AR1)

**We thank the editor for giving us the opportunity to resubmit in addition to the helpful suggestions provided below. We have included responses to each point in addition to updated sections from the manuscript where necessary.**

**In response to Reviewer 1's comment, we are now including results from PMIP4 LGM simulations. Since the PMIP4 LGM data is relatively new and the sea-ice data has not been published yet, we had to directly ask modeling groups for their data. To acknowledge their contributions to our revised manuscript, we therefore invited these authors to be co-authors of our manuscript. As a result, we have now added five co-authors: Deepak Chandan, Gerrit Lohmann, W. Richard Peltier, Xiaoxu Shi and Jiang Zhu.**

EDITOR COMMENTS on cp-2020-155, Green et al. "Evaluating seasonal sea-ice cover over the Southern Ocean from the Last Glacial Maximum" submitted 03 Dec 2020

The authors have put careful thought into addressing the comments of 3 reviewers and one posted comment. Having read their responses and noted the positive inclusion of the new, PMIP4 analysis, I think that the authors are in a good position to move forward to "reconsider after major revisions." In addition to the well-documented changes planned by the authors, I have noted the following for the authors to consider in their final manuscript revision:

1) The authors have indicated that one of the goals of this manuscript is to provide updated LGM proxy datasets. I suggest that the authors could better document in their methods section how they harmonized chronologies (what was the choice of LGM time period), methodologies, and determined consistencies (if needed).

While we were first to submit this dataset in our original manuscript, the dataset has since then been published by Lhardy et al. (2021). Due to this, we can no longer claim to provide an unpublished, updated LGM sea-ice proxy dataset. However, some of the SST data we use was not included in Lhardy et al. (2021). We have clarified this now in the second paragraph of our introduction, lines 45-49:

"Since 2005, additional SO sea-ice data has been published (Allen et al., 2011; Benz et al., 2016; Ferry et al., 2015; Ghadi et al., 2020; Nair et al., 2019; Xiao et al., 2016), and recently merged into an updated compilation (Lhardy et al., 2021). Within this updated sea-ice compilation, certain cores also contain summer SST estimates. We use this sea-ice proxy data along with the summer SST proxy data to better constrain the minimum and maximum LGM sea-ice cover."

In addition, we would like to provide some information on the updated datasets here just for your reference.

The new compilation is mainly based on two large regional datasets: Gersonde et al. (2005) and Benz et al. (2016), which both follow Gersonde et al. (2005) rules for quality levels in the chronologies and

estimations. The quality levels are all reported in these papers. For the additional datasets (Allen 11, Ferry 15, Nair 19, Ghadi 20, Xiao 20): Regarding their stratigraphic quality levels, most use either radiocarbon dating complemented with d18O stratigraphy, biostratigraphy or magnetic susceptibility. They all fall at stratigraphic quality levels better than 3. Chronologies were not harmonized as most cores present 14C-based chronologies with recent calibrations. This should ensure consistency between the cores. Regarding estimation quality levels, the mean winter sea-ice (WSI), summer sea ice (SSI), *Fragilariopsis curta+cyclindrus* (*FCC*), *F. obliquecostata* (*Fobliq*) and sea-surface temperature (SST) were calculated on the EPILOG timeslice (19-23 ka) to follow on Gersonde et al. (2005) recommendations. The quality of the mean obviously depended on the number of samples present in the timeslice. We believe all cores presented several points over the EPILOG timeslice. Quantitative estimates had good dissimilarity or communality, thus being assigned high-quality levels of 1-2 over 3. Qualitative estimates (*FCC* and *Fobliq*) are assigned a lower community level of 3 over 3.

2) Similarly, as the LGM proxy data set update was an identified goal, the authors should devote at least one paragraph at the outset of their Results section outlining what the new compilation shows (prior to their comparison to simulations). Such a paragraph could also help to address concerns of R2 regarding zonal asymmetry. How prominent is zonal asymmetry in WSI cover for both time periods in the proxy data?

Again, we no longer include an updated LGM sea ice dataset within our goals. Nevertheless, please find below some information for your reference.

Overall there are no great changes in winter sea-ice (WSI) extent between the new compilation (Lhardy et al. 2021) and the old one (Gersonde et al., 2005), despite WSI being a bit more expanded in the Scotia Sea (Allen et al., 11). There was more WSI in the Atlantic (off the Weddell Sea) and the Pacific (off the Ross Sea) than in the Indian and eastern Pacific. This is visible in the distance between the WSI edge and the continent (~20° of latitude at 0° of longitude off Weddell Sea; 17-18° of latitude at 150°W off Ross Sea; 13-15° of latitude at 90°E in the Indian sector; ~11° of latitude at 90°W in the western Pacific). Summer sea-ice (SSI) extent would have been the same if restricted to the control points.

3) R1's comments about the feedbacks between winds and sea-ice are important ones. In their response, the authors do a thorough job of clarifying that the goal of the paper is not to address (a) how accurately the models reproduce winds, or (b) how sea-ice feedbacks ultimately influence wind positions. Rather, the authors are focussing on each model's relationships between winds and sea ice.

The first point (a) appears to be well-addressed by the proposed text for Results section 3.4 (below), but the additional sentences added to Results section 3.3 to address point (b) (also) below jumps topics and does not appear positioned well. I suggest adding this response to paragraph just above section 3.2 (ca. line 155 in the original manuscript), and to state it more clearly as "we recognize that the presence or absence of sea ice also has a direct influence on surface winds (Kidston et al. 2011; Sime et al. 2016). However we are focused on the influence of winds on sea ice…."

TEXT FROM AUTHORS: Results section 3.4:
"While latitudinal position and magnitude of southern hemispheric westerlies at the LGM is poorly constrained (Kohfeld et al., 2013; Sime et al., 2016), here we want to assess the impact of the simulated windstress curl on ocean dynamics in each model. We thus use the simulated windstress outputs to estimate the location and strength of the SO upwelling, and its potential impact on sea-ice cover. "

TEXT FROM AUTHORS: "Results section 3.3:
"The strength and location of the southern hemispheric westerly and polar easterly winds impact Southern Ocean circulation, sea ice transport and therefore sea-ice distribution (Purich et al., 2016; Holland and Kwok, 2012). On the other hand, the presence or absence of sea ice also has a direct influence on surface winds (Kidston et al. 2011; Sime et al. 2016). The divergence created by the wind stress curl over the Southern Ocean leads to an upwelling of warmer deep waters and thus heat loss to the atmosphere. This upwelling can therefore also impact Southern Ocean sea-ice distribution."

We thank the editor for these helpful suggestions to clarify our goal of the paper in relation to winds and sea ice. Since we have restructured our results, we decided to include both of the above "TEXT FROM AUTHORS" in the first paragraph of our results section 3.4-Drivers of inter-model variability. This paragraph is shown below:

Lines 265-273:

"The strength and location of the southern hemispheric westerly and polar easterly winds impact Southern Ocean circulation, sea ice transport and therefore sea-ice distribution (Purich et al., 2016; Holland and Kwok, 2012). On the other hand, the presence or absence of sea ice also has a direct influence on surface winds (Kidston et al., 2011;  Sime et al., 2016). Here, we are focused on the influence of winds on sea ice through the divergence created by the wind stress curl. Within the Southern Ocean divergence leads to upwelling of relatively warm circumpolar deep waters and thus heat loss to the atmosphere. This upwelling can therefore also impact Southern  Ocean sea-ice distribution. While the latitudinal position and magnitude of southern hemispheric westerlies at the LGM is poorly constrained (Kohfeld et al., 2013; Sime et al., 2016), we want to assess the impact of the simulated windstress curl on ocean dynamics in each model. We thus use the simulated windstress outputs to estimate the location and strength of the SO upwelling, and its potential impact on sea-ice cover."

4) R2 is concerned that zonal averages will smooth out zonal asymmetries in the sea-ice edge, but authors are unclear on a better way to conduct this analysis. Note that when using paleo-data we get around issues of zonal asymmetry by using a feature like the modern-day polar front and expressing changes in variables (e.g. LGM minus PI) against location, where location is expressed as latitudinal difference from the modern-day front. This would be one approach for addressing this concern. Understandably, although this approach may be very useful, this new analysis may be beyond the scope of this manuscript at this stage. However, it would be useful for authors to address this limitation in the manuscript by (a) describing any zonal asymmetries observed in the paleo-data and

(b) explaining that their analysis does not address how these asymmetries can affect the overall (zonally averaged) estimates of summer and winter sea-ice extent.

We understand this concern and appreciate the suggestion. As you mentioned, at this late stage of the manuscript we do not plan to conduct any more significant additions to the analysis that isn't already presented. However, we now include a paragraph in our discussion that highlights these zonal asymmetries in the proxy data and the implications that they have on our LGM summer sea-ice estimate.

Discussion lines 317-322:

"Our estimate for the LGM SSI edge is a zonally averaged estimate and therefore assumes a fairly circular SSI distribution, similar to that simulated by AWI-ESM-1 and FGOALS-G2 (Figure 1). While the LGM SSI proxy data is limited, Lhardy et al. (2021) suggest the three basins behaved very differently, with a LGM SSI edge at 54°S in the Atlantic, 65-66°S in the Indian, 63°S in the western Pacific and 66-68°S in the eastern Pacific. If this indeed was the case, our suggested LGM SSI edge would potentially overestimate the sea ice edge in some regions while potentially underestimating it in other regions. Additional proxy data from the Pacific and Indian basins would reduce the uncertainty of our estimate."

5) Please include latitude/longitude grid lines in your new figures so that readers can see latitudinal positions of data and modeled sea ice extents (e.g. Figure 1).

We appreciate this suggestion and have done this.

6) The authors might better address R2's comment about regridding of model simulations by simply indicated in figure captions where this has occurred, or putting one sentence in the methods.

We have decided to keep all figures and tables based on the regridded data and due to this, we chose not to include this information in every figure caption. In hopes of clarifying the method of interpolation for Reviewer 2, we have adjusted the following sentence to section 2.1 of the Methods, lines 101-102:

"To ease the comparison, we used bilinear interpolation to standardize each model to a 1˚x 1˚grid with the CDO software (Climate Data Operators, Schulzweida et al. 2014)."

---

## Author Response (AR2)

Dear Dr. Kohfeld,

Thank you very much for accepting our manuscript for submission. We appreciate you catching the typo related to Figure 3 and your other helpful suggestions that have improved the clarity of our manuscript. Below we state how we addressed each comment in blue.

LN 175-176: for clarity, please change "extending past" to "extending equatorward of" to indicate direction of change in sea ice extent
This has been changed.

LN 247-249: the definitions of proxy/model lines appear to be reversed in the text (proxy should be solid black line; model dotted) relative to figure 3.
Thank you for pointing this out, you are correct. It has now been fixed.

LN 348-349: For clarity, can you please specify that the relationships are with SSI, rather than the more general use of "sea-ice extent"?
To make the sentences clearer, we have replaced sea-ice extent with SSI extent, shown below:

Lines 345-358: "There is however a weak relationship between SSI extent and AMOC depth (Fig. S2, $R2=0.17$), with a shallower AMOC generally associated with a larger SSI extent. A larger SSI extent, and thus increased sea-ice formation, could impact the AABW properties and therefore ocean stratification (Marzocchi and Jansen, 2017), as evident from Fig. 5."

LN 358: To make your sentence parallel, can you please also provide the approximate, seasonal range of the sea-ice extent in degrees latitude for the modern day as well?
Estimated from Fig. 1 of Cavalieri and Parkinson (2012), we have now included an approximate present-day sea-ice edge seasonality ranging from ~15˚ in the Atlantic sector and less than 5˚ in the Indian sector. This sentence is shown below:

Lines 355-356: "In comparison, the present day seasonal change in sea-ice edge ranges from ~15˚ in the Atlantic sector to less than 5˚ in the Indian sector (Cavalieri and Parkinson, 2012)."